# Contrasting the Penultimate and Last Glacial Maxima (140 and 21 ka BP) using coupled climate-ice sheet modelling

Violet L. Patterson[1], Lauren J. Gregoire[1], Ruza F. Ivanovic[1], Niall Gandy[2], Jonathan Owen[1], Robin S. Smith[3], Oliver G. Pollard[1], Lachlan C. Astfalck[4], Paul J. Valdes[5]

[1]School of Earth and Environment, University of Leeds, Leeds, UK
[2]Department of the Natural and Built Environment, Sheffield Hallam University, Sheffield, UK
[3]NCAS, Department of Meteorology, University of Reading, Reading, UK
[4]School of Physics, Mathematics and Computing, University of Western Australia, Perth, Australia
[5]School of Geographical Sciences, University of Bristol, Bristol, UK

*Correspondence to*: Violet L. Patterson (ee17vp@leeds.ac.uk)

**Abstract.** The configuration of the Northern Hemisphere ice sheets during the Penultimate Glacial Maximum differed to the Last Glacial Maximum. However, the reasons for this are not yet fully understood. These differences likely contributed to the varied deglaciation pathways experienced following the glacial maxima and may have had consequences for the interglacial sea level rise. To understand the differences between the North American Ice Sheet at the Last and Penultimate Glacial Maxima (21 and 140 ka BP), we perform two perturbed-physics ensembles of 62 simulations using a coupled atmosphere-ice sheet model, FAMOUS-ice, with prescribed surface ocean conditions, in which the North American and Greenland ice sheets are dynamically simulated with the Glimmer ice sheet model. We apply an implausibility metric to find ensemble members that match reconstructed ice extent and volumes at the Last and Penultimate Glacial Maxima. We use a resulting set of 'plausible' parameters to perform sensitivity experiments to decompose the role of climate forcings (orbit, greenhouse gases) and initial conditions on the final ice sheet configurations. This confirms that the initial ice sheet conditions used in the model are extremely important in determining the difference in final ice volumes between both periods due to the large effect of the ice-albedo feedback. In contrast to evidence of a smaller Penultimate North American Ice Sheet, our results show that the climate boundary conditions at these glacial maxima, if considered in isolation, imply a larger Penultimate Glacial Maximum North American Ice Sheet than at the Last Glacial Maximum, of around 6 meters sea level equivalent. This supports the notion that the growth of the ice sheet prior to the glacial maxima is key in explaining the differences in North American ice volume.

## 1 Introduction

The Penultimate Glacial Maximum (PGM) occurred around 140,000 years ago, within Marine Isotope Stage 6 (MIS 6). Greenhouse gas (GHG) concentrations and global average insolation were similar to the Last Glacial Maximum (LGM; ~21 ka BP) (Berger and Loutre, 1991; Loulergue et al., 2008; Bereiter et al., 2015) but the orbital configuration differed, affecting the seasonal and latitudinal distribution of incoming shortwave radiation (Berger, 1978; Colleoni et al., 2011). The global total ice sheet volume, and thus the global mean sea level, was likely similar between the two glacial maxima (~120-130 m below

present), with larger uncertainty at the PGM (Rabineau et al., 2006; Masson-Delmotte et al., 2010; Rohling et al., 2017). Both geological evidence and numerical modelling suggest that despite the similarities in total ice volume between the PGM and the LGM, the configurations of the Northern Hemisphere ice sheets differed significantly (e.g. Svendsen et al., 2004; Colleoni et al., 2016; Batchelor et al., 2019).

Some reconstructions suggest the Eurasian Ice Sheet (EIS) may have been up to ~50 % larger during the Penultimate Glacial Cycle (MIS 6: ~190-130 ka BP) than during the Last Glacial Cycle (~115-12 ka BP) (Svendsen et al., 2004). However, evidence of multiple advances and uncertainties in dating proxy records means that the maximum extent mapped at 140 ka BP could correspond to previous advances during MIS 6. Similarly, the timing of the maximum extent of the EIS at the LGM is also uncertain and areas of the ice margin likely reached their maximum extents at different times throughout the glacial cycle (Svendsen et al., 2004; Margari et al., 2014; Colleoni et al., 2016; Ehlers et al., 2018). The extent of the North American Ice Sheet (NAIS) during the PGM is even less well constrained due to a lack of glaciological evidence (e.g. moraines and till). The scarcity of empirical data in itself suggests that it was smaller in most areas than at the LGM because the subsequent larger ice sheet could have largely erased the evidence of prior glaciations (Dyke et al., 2002; Rohling et al., 2017). Additionally, evidence of reduced ice rafted debris (IRD) discharge from the Hudson Strait in the North Atlantic IRD belt (e.g. Hemming, 2004; Naafs et al., 2013; Obrochta et al., 2014), relative sea level assessment studies (e.g. Rohling et al., 2017) and climate, ice sheet and glacial isostatic adjustment modelling (e.g. Colleoni et al., 2016; Dyer et al., 2021) all point to a smaller volume PGM NAIS. For example, assuming a similar global mean sea level fall (and Antarctic ice sheet volume) at the PGM as at the LGM but with a larger volume EIS at the PGM (estimated at 33-53 m sea level equivalent (SLE) versus 14-29 m SLE at the LGM), this follows that the NAIS must have been smaller than at the LGM to compensate (39-59 m SLE versus 51-88 m SLE) (Rohling et al., 2017).

The reason for these differences is likely complex and is not yet fully understood. The evolution and surface mass balance (SMB) of ice sheets depends on many factors such as; background climate, climate and ice sheet histories, dust deposition, vegetation, ice albedo and sea surface temperatures, as well as the interactions and feedbacks between them all (Kageyama et al., 2004; Krinner et al., 2006; 2011; Colleoni et al., 2009a; 2011; Liakka et al., 2012; Stone and Lunt, 2013). The ice sheets themselves also strongly influence the climate through their interactions with atmospheric and oceanic circulation and the energy balance. This alters global and local temperature and precipitation patterns which in turn affects ice sheet ablation and accumulation (i.e. SMB) (e.g. Kageyama and Valdes, 2000; Abe-Ouchi et al., 2007; Beghin et al., 2014; 2015; Ullman et al., 2014; Liakka et al., 2016; Gregoire et al., 2015; 2018; Snoll et al., 2022; Izumi et al., 2023). These interactions between the vast ice sheets and other components of the climate system exerted an important control on the initial climate state for the deglaciations, and hence on the subsequent chain of events, thus impacting the climate, ocean and sea level evolution during deglaciation. Thus, the contrasting configurations of the Northern Hemisphere ice sheets at the glacial maxima may have contributed to the different deglaciation pathways that followed. In this context, it is important to examine the complex physical interactions between the climate and the ice sheets to better understand why the last two glacial maxima had different ice sheet

configurations and evaluate the ice sheets' sensitivities to changes in climate in relation to different orbits and greenhouse gas concentrations. To achieve this, numerical simulations of these periods are required using a coupled climate-ice sheet model that capture these complex, non-linear interactions. Previous studies on glacial-interglacial cycles, have relied on the coupling of relatively fast, low resolution and simplified Earth system Models of Intermediate Complexity (EMICs) to an ISM (e.g. Bonelli et al., 2009; Ganopolski et al., 2010; Fyke et al., 2011; Heinemann et al., 2014; Beghin et al., 2014; Ganopolski and Brovkin, 2017; Quiquet et al., 2021; Poppelmeier et al., 2023; Willeit et al., 2024) or one-way forcing of an ice sheet model with climate forcing output by stand-alone climate simulations (e.g. Abe-Ouchi et al., 2013; Stone and Lunt, 2013; Gregoire et al., 2015; 2016). These computationally efficient techniques advanced our understanding of the roles of orbit and $CO_2$ in ice sheet evolution and proposed plausible reconstructions of past ice sheets (e.g. Robinson et al., 2011; Stone et al, 2013). They also highlighted important earth system interactions (e.g. Stone and Lunt, 2013; Willeit et al., 2024) such as with vegetation, dust, albedo, glacial isostatic adjustment, disparate ice sheets (Beghin et al., 2015) as well as internal ice sheet instabilities (Gregoire et al., 2012; Quiquet et al., 2021). However, the accuracy of these results has been limited by the simplified representation of climate processes, atmospheric circulation and/or surface mass balance. A combination of increased computer power, the development of more computationally efficient, lower resolution General Circulation Models (GCMs) and sub-grid scale schemes translating ice sheet relevant atmospheric processes onto the higher resolution ice sheet grid, has made bi-directional, coupled climate-ice sheet simulations over longer timescales, and in large ensembles, feasible (Vizcaino et al., 2013; Ziemen et al., 2014; Sellevold et al., 2019; Smith et al., 2021). These coupled models have been used to simulate the climate-ice sheet interactions during past glacial periods including; glacial inception (Gregory et al., 2012); the LGM and the build up to it (Ziemen et al., 2014; Gandy et al., 2023; Sherriff-Tadano et al., 2023; Nui et al., 2024) and MIS 13 (Niu et al., 2021).

To better understand the differences between the Penultimate and Last Glacial Maxima ice sheet configurations, we seek to establish how the differences in climate forcings (such as orbit and greenhouse gases) between the two periods affected ice sheet surface mass balance and in turn their geometry. To this end, this study uses a coupled atmosphere-ice sheet model (FAMOUS-ice; Smith et al., 2021), to perform ensembles of simulations of the PGM and LGM to explore input climate and ice sheet parameter uncertainties and their effects on the North American ice sheet volume during each period. We identify simulations that match volume and extent constraints and use these to perform a factorial decomposition of the effects of climate forcing and initial conditions on ice volume difference between the two Glacial Maxima.

**2 Methods**

**2.1 Model description**

FAMOUS is a fast, low resolution AOGCM that is based on Hadley Centre coupled model HadCM3 and therefore retains all the complex processes represented in an AOGCM but uses only half the spatial resolution and a longer time step. Since it

requires only 10 % of the computational costs of HadCM3, it has been successfully used for long transient palaeo simulations (Smith and Gregory, 2012; Gregory et al., 2012; Gregoire et al., 2012; Roberts et al., 2014; Dentith et al., 2019) and large ensembles for uncertainty quantification (Gregoire, 2010; Gandy et al., 2023). This study uses the atmospheric component, which is a hydrostatic, primitive equation grid point model with a horizontal resolution of 7.5° longitude by 5° latitude with 11 vertical levels and a 1-hour time step (Williams et al., 2013). Land processes are modelled using the MOSES2.2 land surface scheme (Essery et al., 2003), which uses a set of sub-gridscale tiles in each grid box to represent fractions of nine different surface types, including land ice (Smith et al., 2021). Whilst this study prescribes sea surface temperatures and sea ice concentrations, FAMOUS can also be run fully coupled with a dynamical ocean (e.g. Dentith et al., 2019).

FAMOUS now allows the direct two-way coupling to an ice sheet model in the configuration FAMOUS-ice (Smith et al., 2021). Here, we use FAMOUS in combination with Glimmer to interactively simulate the North American and Greenland ice sheets at 40km resolution. Glimmer is a fast running, 3D thermomechanical ice sheet model which uses the shallow ice approximation. This allows it to model ice sheet evolution over long timescales as it is more computationally efficient, and therefore has been used to simulate continental ice sheets over glacial-interglacial cycles (Rutt et al., 2009; Gregoire et al., 2016). The internal ice temperature is resolved over all 11 layers of the ice sheet and allowed to evolve throughout the simulations under the influence of heat conduction, internal friction and ice advection. The surface temperature boundary condition is set equal to the annual mean surface air temperature, up to a maximum of 0°C and the basal temperature is controlled by the geothermal heat flux (set to $-0.05$ W m$^{-2}$) and friction from sliding (Rutt et al., 2009).

FAMOUS-ice accounts for the mismatch between atmosphere and ice sheet grid sizes by using a multilayer surface snow scheme to calculate SMB on 'tiles' at 10 set elevations within each grid box that contains land ice in FAMOUS. This SMB is then downscaled from the coarse FAMOUS grid to the much finer Glimmer grid at each model year (Smith et al., 2021). Glimmer uses this SMB field to calculate ice flow and surface elevation and passes this back to FAMOUS in which orography and ice cover is updated. In this study, to reduce computational costs further, FAMOUS-ice runs at 10 times ice sheet acceleration: for every year of climate integrated in FAMOUS, the simulated SMB field forces 10 years of ice sheet integration in Glimmer. Figure 1 shows a simplified diagram of this coupling process and full details can be found in Smith et al., (2021). The current computational cost of this set up is around 50 decades (of climate years) per wallclock day using 8 processors (~ 192 core hours).

FAMOUS-ice has been shown to perform well in simulations of past and future ice sheets including Greenland and North America (Gregory et al., 2020; Smith et al., 2021; Gandy et al., 2023). In particular, the LGM North American Ice Sheet study of Gandy et al., (2023) was able to utilise the useful constraints of the LGM to infer the importance of parameters controlling ice sheet albedo on ice sheet configuration in this model.

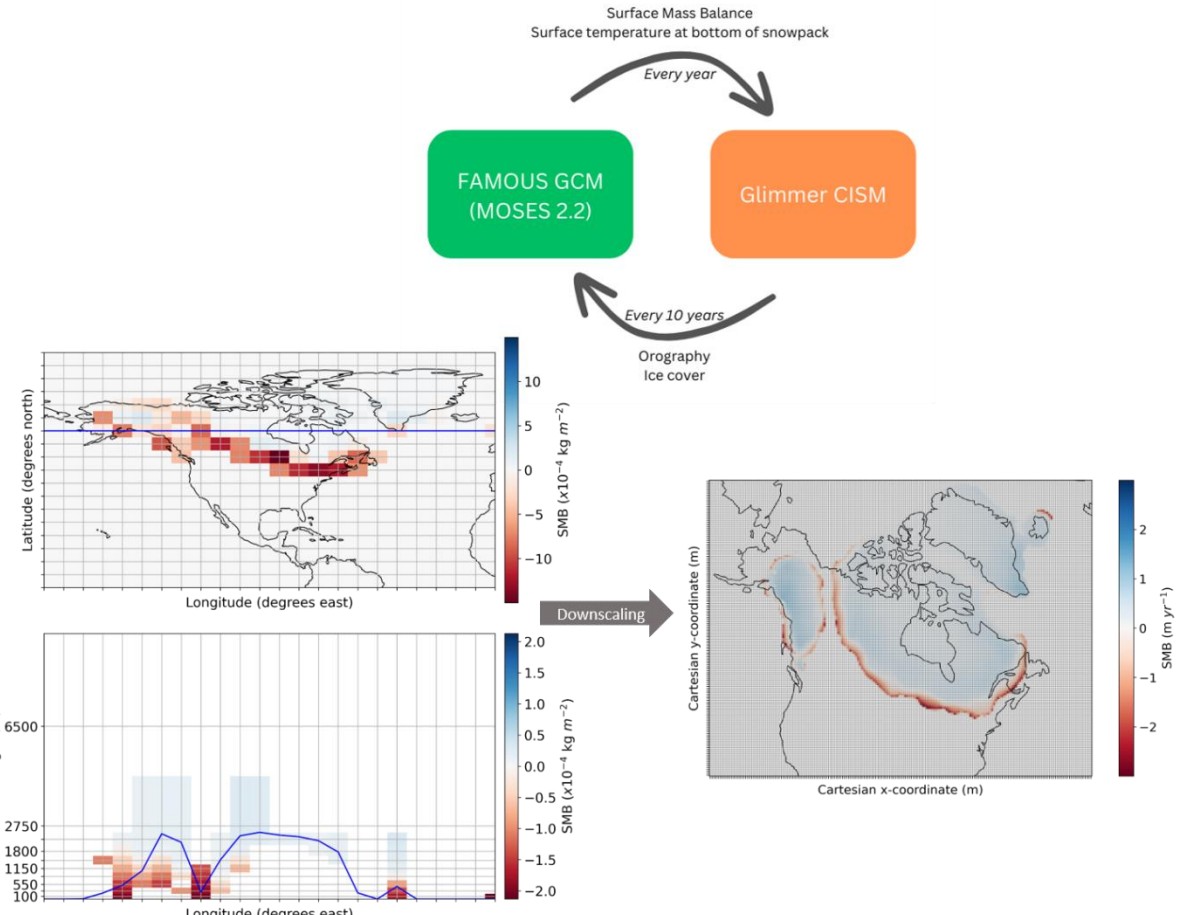

**Figure 1. Schematic illustrating the calculation of SMB along a specific transect across the ice sheet (blue line) at different elevations on the FAMOUS grid followed by downscaling onto the Glimmer grid.**

## 2.2 Experiment design

### 2.2.1 Climate boundary conditions

With the exception of including dynamic North American and Greenland ice sheets, our FAMOUS-ice simulations are set up following the Paleoclimate Modelling Intercomparison Project Phase 4 (PMIP4) protocols for the LGM (Kageyama et al., 2017) and PGM (Menviel et al., 2019). These protocols prescribe climatic boundary conditions, including orbital parameters and GHG concentrations, the values of which can be found in Table 1. Concentrations of $CO_2$, $CH_4$ and $N_2O$ are very similar between the LGM and PGM but orbital parameters are significantly different. The larger eccentricity at the PGM enhances the effect of precession compared to the LGM which affects the seasonal and latitudinal distribution of insolation. These changes are important for ice sheet surface mass balance since melting is particularly sensitive to spring and summer temperatures

(Huybers, 2006; Niu et al., 2019). The PGM received lower insolation in the Northern Hemisphere in late winter to early summer but higher levels in late summer to early winter, compared to the LGM (Fig. 2a). Subsequent to the completion of this work, it was discovered that the equation for the role of eccentricity on solar insolation was incorrect in the model code. The magnitude of the error is larger for periods with higher eccentricity values and so a sensitivity test was run to determine the effect this correction has on SMB and ice volume at the PGM. Details of this error and the results of the sensitivity test can be found in Appendix A, but the impact was shown to be minimal (Fig. A1).

**Table 1. Climate boundary conditions used in the LGM and PGM experiments as prescribed by the PMIP4 protocols for each period (Kageyama et al., 2017; Menviel et al., 2019).**

| | *Eccentricity* | *Obliquity (°)* | *Perihelion – 180 (°)* | *Solar Constant (Wm$^{-2}$)* | *CO$_2$ (ppm)* | *CH$_4$ (ppb)* | *N$_2$O (ppb)* | *Orography and ice extent* |
|---|---|---|---|---|---|---|---|---|
| *LGM (21 ka)* | 0.019 | 22.949 | 114 | 1360.7 | 190 | 375 | 200 | GLAC-1D (Tarasov et al., 2012; Briggs et al., 2014; Ivanovic et al., 2016) |
| *PGM (140 ka)* | 0.033 | 23.414 | 73 | 1360.7 | 191 | 385 | 201 | Combined reconstruction (Abe-Ouchi et al 2013; Briggs et al 2014; Tarasov et al 2012) |

In the climate model, the global orography (including the Eurasian and Antarctic ice sheets) and land-sea mask for the LGM are calculated from the GLAC-1D 21 ka BP reconstruction (Tarasov et al., 2012; Briggs et al., 2014; Ivanovic et al., 2016), which is one of three recommendations in the PMIP4 protocol (Kageyama et al., 2017). For the PGM simulations we used the 140 ka BP combined reconstruction (Tarasov et al., 2012; Abe-Ouchi et al., 2013; Briggs et al., 2014) detailed in the PGM PMIP4 protocol (Menviel et al., 2019). Vegetation is prescribed based on a pre-industrial distribution and kept constant. As ice cover changes, the fractions of grid cells that are land ice versus other surface types changes proportionally, altering albedo. However, since there is no dynamical vegetation component, some important climate-ice-vegetation feedbacks are neglected, which could have a significant impact on ice sheet evolution (Stone and Lunt, 2013).

Because of the low resolution of the FAMOUS model, using a dynamical ocean and sea ice can introduce large biases in the simulated climate (Dentith et al. 2019). By prescribing Sea Surface Temperature (SST) and sea ice, we are able to limit the amplification of climate biases arising from atmosphere-ocean-sea ice interactions. Thus, SSTs and sea ice concentration are also prescribed and constant and are taken from higher resolution HadCM3 simulations of 21 ka BP (Fig. B1a; see details in Izumi et al., 2022) and 140 ka BP (Fig. B1b). The 140 ka BP simulation is part of a suite of simulations covering the last 140,000 years (Allen et al., 2020). It was performed using a version of HadCM3 (specifically HadCM3B-M2.1aD, see Valdes et al., (2017), which was the same version as used by Izumi et al., (2022) for the LGM and Davies-Barnard et al., (2017)). The simulation was forced with 140 ka BP orbital configuration (Berger and Loutre, 1991) and greenhouse gases (Petit et al., 1999; Spahni et al., 2005; Loulergue et al., 2008). Ice sheet forcing and land sea mask were from DeBoer et al., (2013) who modelled the evolution of all the major ice sheets. It was run as a "snapshot" simulation for 3070 years which allowed the deeper ocean to attain near equilibrium.

FAMOUS atmosphere-ocean GCM has not been run for the PGM, and we lack sufficient data density for precisely dated PGM SSTs and sea-ice to produce statistically varied reconstructions, as in Gandy et al., (2023). Thus, for physical consistency between the LGM and PGM periods, HadCM3 output was used for the surface ocean boundary conditions. Of all possible options, HadCM3 output is the most appropriate choice for this because it is the parent model for FAMOUS; they share the same physics, differing mainly in their resolutions, and HadCM3 was used as the tuning target for FAMOUS during model development (Smith et al., 2008). We take the multi-year monthly mean "climatology" of SSTs and sea ice concentrations from the final 100 years of the simulations. These 12-month climatologies are repeated throughout the duration of the simulations to provide a seasonal forcing with no long-term trend and no interannual variability.

The modelled annual average SSTs are cooler at the LGM than at the PGM, everywhere, except in the North Atlantic due to less sea ice cover in this region (Fig. 2b). However, the summer SSTs are warmer in the Northern Hemisphere at the LGM compared to the PGM (Fig. 2c). The HadCM3 LGM SSTs are colder on average than the reconstruction in Gandy et al., (2023), with the largest differences, of up to 6 °C, occurring in the tropics and mid-latitudes (Fig. B1c).

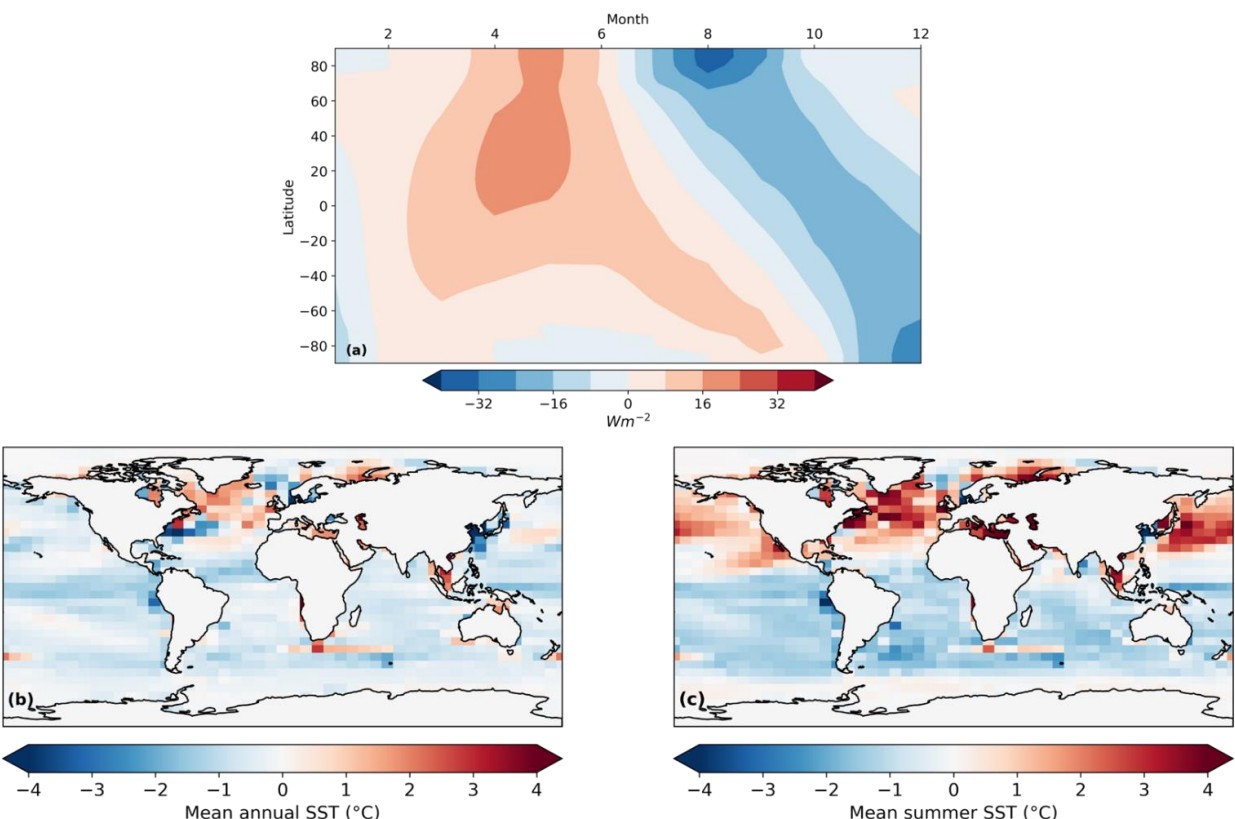

**Figure 2. Difference between the LGM and PGM (a) latitudinal distribution of incoming top of the atmosphere shortwave radiation each month (b) modelled annual sea surface temperatures and (c) modelled summer (JJA) sea surface temperatures.**

### 2.2.2 Ice sheet boundary and initial conditions

In all our simulations, the ice sheet extent is set to the PMIP4 boundary conditions for the LGM and PGM as described in Table 1, except in the interactive ice sheet model domain, which covers North America and Greenland. Here, we describe how the ice extent and elevation is initialised in FAMOUS and Glimmer over the interactive domain in our ensemble of PGM and LGM simulations and sensitivity experiments.

In our ensemble of LGM and PGM simulations, Glimmer is initiated from an 18.2 ka BP NAIS taken from a previous Last Deglaciation ensemble (Gregoire et al., 2016). This smaller intermediate (MIS 3-like) ice sheet was used in Gandy et al., (2023) as an approximate pre-glacial maximum extent from which to grow the ice sheet towards an equilibrium ice volume. For consistency, we used the same initial ice sheet conditions as in Gandy et al. (2023) when running our ensembles of LGM and PGM simulations. The coupling between the models passes this orography field from Glimmer to FAMOUS, updating the PMIP4 boundary condition that FAMOUS was initiated from. However, due to the technical formulation of the coupling, where entire gridboxes were initialised as covered in ice at all elevations in FAMOUS, the tiles in such gridboxes would not

subsequently update to reflect the existence of any non-glaciated fractions that might exist in the Glimmer state. This means that when the initial conditions are radically different in FAMOUS and Glimmer (as in our ensemble of simulations), the FAMOUS ice extent over the North American continent is not updated to match the Glimmer initial conditions. Thus, in our ensemble of LGM simulations, the albedo remains high throughout the saddle region (the area between the Laurentide and Cordilleran ice sheets) because the FAMOUS ice extent remains as large as the atmospheric model's initial conditions (i.e. the GLAC-1D 21 ka BP reconstruction) for the duration of the simulations (Fig. 3). This coupling procedure has since been improved to allow tile fractions to update to match those in the ice sheet model despite drastically different initial ice cover. The different ice sheet configurations used in FAMOUS and Glimmer in the ensembles, are outlined in our table of experiments, Table 2 (experiments 1 and 2). The impact of this set-up compared to an ice sheet configuration matched in FAMOUS and Glimmer is explored in Sect. 3.2 and Appendix C.

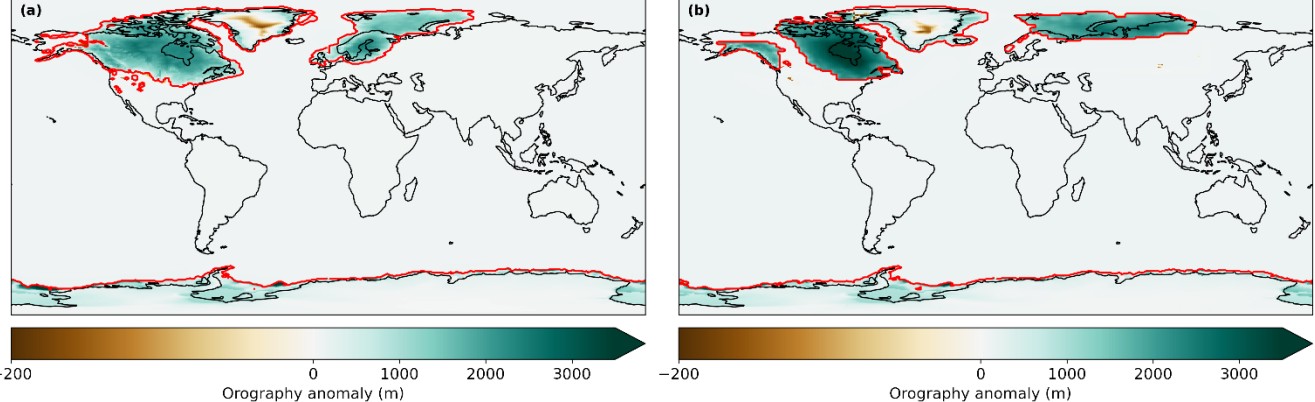

**Figure 3. Topography anomaly from present day used as the initial condition in FAMOUS and the ice masks (red lines) for (a) the LGM and (b) the PGM.**

We perform two sets of sensitivity experiments to understand the relative impact of the initial ice sheet conditions and the climate forcing on the resulting LGM and PGM NAIS volumes. The first set of experiments uses matching ice sheet configurations in FAMOUS and Glimmer, set either to the LGM GLAC-1D reconstruction or to the end of one of our PGM coupled simulations (Table 2; experiments 3 – 6). The second set uses the same initial ice sheet configurations as in the ensemble, i.e. GLAC-1D and PMIP4 reconstructions in FAMOUS and the 18.2 ka ice sheet in Glimmer (Table 2; experiments 7 - 10). A full description of the initial conditions and methods used in these sensitivity experiments can be found in Sect. 2.5.

**Table 2. Table of experiments performed in this study detailing the 'climate forcing' (orbital configuration, trace gases and global orography as outlined in Table 1 and SSTs/sea ice from HadCM3), initial ice extent set in FAMOUS over Greenland and North America, initial Glimmer ice sheet conditions and input parameter values. NROY are the simulations that are 'Not Ruled Out Yet' after applying the implausibility metric described in Sect. 2.4.**

| Experiments | Climate forcing | FAMOUS initial ice extent | Glimmer initial condition | Input parameter values |
|---|---|---|---|---|
| 1) LGM ensemble | LGM | PMIP4 LGM (GLAC-1D) | 18.2 ka ice sheet | Randomly sampled from Table 3 ranges (See Sect. 2.3) |
| 2) PGM ensemble | **PGM** | **PMIP4 PGM** | 18.2 ka ice sheet | Randomly sampled from Table 3 ranges (See Sect. 2.3) |
| 3) $V_{\_1}$ (full LGM) | LGM | PMIP4 LGM (GLAC-1D) | PMIP4 LGM GLAC-1D | Matching NROYa simulation xpken/xpkyn (See Sect. 2.4 and 3.1) |
| 4) $V_{c\_1}$ | **PGM** | PMIP4 LGM (GLAC-1D) | PMIP4 LGM GLAC-1D | |
| 5) $V_{i\_1}$ | LGM | **PGM NROYa (xpkyn)** | **PGM NROYa (xpkyn)** | |
| 6) $V_{ci\_1}$ (full PGM) | **PGM** | **PGM NROYa (xpkyn)** | **PGM NROYa (xpkyn)** | |
| 7) $V_{\_2}$ (NROYa LGM) | LGM | PMIP4 LGM (GLAC-1D) | 18.2 ka ice sheet | |
| 8) $V_{c\_2}$ | **PGM** | PMIP4 LGM (GLAC-1D) | 18.2 ka ice sheet | |
| 9) $V_{i\_2}$ | LGM | **PMIP4 PGM** | 18.2 ka ice sheet | |
| 10) $V_{ci\_2}$ (NROYa PGM) | **PGM** | **PMIP4 PGM** | 18.2 ka ice sheet | |

## 2.3 Ensemble design

The ensemble by Gandy et al., (2023) showed that uncertainty in parameters controlling SMB, ice sheet dynamics and climatic conditions over the ice sheets had a significant influence on the extent and volume of the LGM NAIS, with albedo parameters explaining the majority of the variation in model output. Since these parameters needed re-tuning from simulations of the present day Greenland ice sheet to produce an acceptable LGM NAIS configuration in FAMOUS-ice under LGM climate conditions, the PGM may also show different sensitivities to the uncertain parameters. Therefore, we ran new ensembles of the LGM and PGM in order to explore uncertainties and identify combinations of climate and ice sheet parameters that perform well for both periods.

Following on from Gandy et al., (2023), a second wave of simulations was performed and compared to reconstructions of ice sheet extent and volume to identify 'Not Ruled Out Yet' (NROY) parameter combinations (see methodology in Appendix D), the results of which formed the basis of the ensemble design in this study. We reran the LGM ensemble to allow for slight changes in the experiment design compared to Gandy et al., (2023): we use orbital parameters for 21 ka BP rather than 23 ka BP and HadCM3 SSTs instead of a statistical reconstruction (see Sect. 2.2.1). Table 3 details the 13 parameters that were

varied in these simulations. Out of the 176 NROY parameter combinations from the Wave 2, a representative subset of 62 were selected which provided adequate coverage of the NROY space (see Appendix D for details). Each was run for 1000 climate years (10,000 ice sheet years) for both the LGM and PGM experiments until the majority of the ice sheet reached close to equilibrium. Despite differences in the model set up between this study and Gandy et al., (2023), we expect the 62 samples chosen from their design to be a good estimate to an optimal parameter design for our experiment design (Appendix D).

**Table 3. Description of parameters varied in the ensembles. Adapted from Gandy et al., (2023).**

| Parameter | Range | Description |
|---|---|---|
| Lapse Rate | $-0.01 - -0.002$ K km$^{-1}$ | Prescribed lapse rate for air temperature used to downscale FAMOUS near-surface ice sheet climate onto surface elevation tiles. Down welling longwave radiation is also adjusted for consistency. More negative values lead to stronger lapse rate effects (Smith et al., 2021). |
| Daice | $-0.4 - 0$ K$^{-1}$ | Sensitivity of bare-ice albedo to surface air temperatures once the surface is in a melt regime. Albedo reduced to as low as 0.15 with minimum value (Smith et al., 2021) |
| Fsnow | $350 - 800$ kg m$^{-3}$ | The threshold in surface snow density at which the FAMOUS albedo scheme switches from a scattering paradigm appropriate for a conglomeration of snow grains to one more appropriate for a solid surface. Higher values correspond to using brighter albedos for denser snow, increasing ice sheet albedo (Smith et al., 2021) |
| AV_GR | $0 - 0.01$ $\mu$ m$^{-1}$ | Sensitivity of the snow albedo to variation in surface grain size. Higher values enhance the darkening of snow over time, decreasing the albedo (Smith et al., 2021). |
| RHcrit | $0.6 - 0.9$ Pa$^{-1}$ | The threshold of relative humidity for cloud formation (R. Smith, 1990). A higher value means clouds can form less easily. |
| VF1 | $1 - 2$ m s$^{-1}$ | The precipitating ice fall-out speed (Heymsfield, 1977). |
| CT | $5\times10^{-5} - 4\times10^{-4}$ s$^{-1}$ | The conversion rate of cloud liquid water droplets to precipitation (R. Smith, 1990). |
| CW | $1\times10^{-4} - 2\times10^{-3}$ kg m$^{-3}$ | The threshold values of cloud liquid water for formation of precipitation (R. Smith, 1990). Only the value for the land is varied. |
| Entrainment Coefficient | $1.5 - 6$ | Rate of mixing between environmental air and convective plume. Higher values enhance mixing of convective plumes with ambient dry air. |
| Alpham | $0.2 - 0.65$ | The sea ice lowest albedo (Crossley & Roberts, 1995). |
| Basal Sliding | $0.5 - 20$ mm yr$^{-1}$ | The basal sliding rate. A higher value allows increased ice velocity. |

| | | |
|---|---|---|
| *Mantle Relaxation Time* | 300 – 9000 yrs | The relaxation time of the mantle, a lower value making the mantle less viscous, thus allowing a quicker topographic rebound. |
| *Flow Enhancement Factor* | 1 - 10 | Glen's Flow Law enhancement factor. Increasing the factor makes the ice softer and more deformable (Rutt et al., 2009). |

240

**2.4 Implausibility criteria**

To filter out implausible ice sheet configurations in the results, a set of constraints, based on southern ice sheet extent and volume, were applied to the LGM ensemble. Both ensembles were filtered based on the LGM results since the extent of the NAIS is very well constrained by geological data and there are more estimates of ice volume for the LGM than the PGM. This is because there is a lack of empirical data (over both space and time) on ice sheet configuration at the PGM due to destruction of evidence by subsequent glaciations and difficulties with dating what is available (Parker et al., 2022). Thus, most of the reconstructions of NAIS PGM extent are actually the maximum extent reached over the whole of MIS 6 (190-132 ka BP) and are mostly based on numerical modelling combined with this scarce proxy data (e.g. Colleoni et al., 2016; Batchelor et al., 2019). This leaves a set of plausible or 'Not Ruled Out Yet' (NROY) LGM simulations that can then be compared to the corresponding PGM simulations to determine whether parameters that performed well for the LGM also give plausible PGM results. LGM ice extent was assessed against the reconstruction by Dalton et al. (2020). We focus our evaluation of ice extent on the southern NAIS area and chose to disregard regions of known model bias. This includes marine margins that are subject to processes not included in Glimmer and the Alaskan regions where small climate model biases lead to ice sheet overgrowth (e.g. Ganopolski et al., 2010; Ziemen et al., 2014; Gregoire et al. 2016, Sherriff-Tadano et al., 2023). Additionally, ice lobes are not well captured in many models as they are likely to be transient, short-lived features that may be caused by complex ice dynamics (e.g. Zweck and Huybrechts, 2005). Therefore, we do not expect our simulations to perfectly match the reconstructed Southern NAIS extent. To account for the expected mismatch between model and data, we applied a tolerance on the Southern ice sheet area of $1.79 \times 10^6$ km$^2$, equivalent to three-times the area of the lobes (Fig. 4). We thus calculate the Southern NAIS ice area as the integrated area within the large box shown in Fig. 4 at the end of each LGM simulation and selected simulations that matched the reconstructed area from Dalton et al. (2020) within plus or minus $1.79 \times 10^6$ km$^2$. The volume of the NAIS is not as well constrained by proxy data and so estimates rely on ice sheet, glacial isostatic adjustment and sea level modelling studies. Based on a number of these studies (Marshall et al., 2002; Tarasov and Peltier, 2002; 2004; Tarasov et al., 2012; Lambeck et al., 2014; Peltier et al., 2015; Rohling et al., 2017; Batchelor et al., 2019; Gowan et al., 2021), a minimum NAIS (including Greenland) volume of 70 m SLE ($2.8 \times 10^7$ km$^3$) was applied to the ensemble. The translation of ice volumes into meters of sea level equivalent are calculated based on present day ocean area.

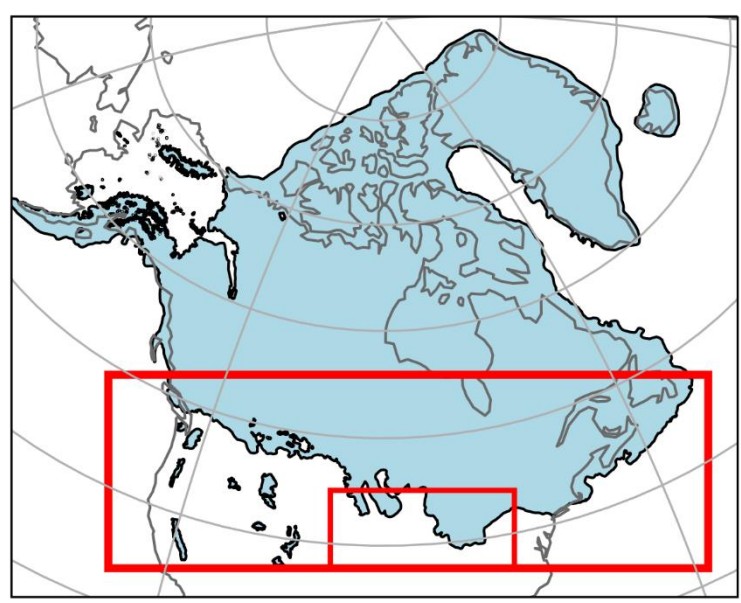


**Figure 4. Outline of the LGM North American Ice Sheet by Dalton et al. (2020). The large red box shows the region used to calculate**
**reconstructed and modelled Southern NAIS area. The small red box shows the region used to calculate the area of the lobes from**
**which we set the upper and lower target bounds for southern ice extent (See Sect. 2.4).**
**2.5 Sensitivity analysis**
We choose one of the resulting NROY parameter combinations, NROYa (specifically experiments xpken/xpkyn), which has
LGM and PGM ice volumes lying in the middle of estimated ranges and the least excess ice growth over Alaska, to investigate
the relative impact of the initial conditions versus the climate on the resulting ice sheet configurations. This is achieved through
a sensitivity analysis along with factorisation based on the method used in Lunt et al., (2012) and Gregoire et al. (2015). We
divided the differences in inputs between LGM and PGM into two factors; the initial ice sheet configurations used in FAMOUS
and Glimmer and the climate boundary conditions (orbital parameters, greenhouse gases and SSTs/sea ice). Thus, the total
difference in final ice volume ($\Delta V$) between the LGM and the PGM can be written as Eq. (1):
$$\Delta V = dV_{ice} + dV_{climate} \,, \tag{1}$$
where $dV_{ice}$ is the difference in final ice volume due to the different initial ice sheet configurations and $dV_{climate}$ is the
difference due to the difference climate boundary conditions used.
The factorisation method requires $2^N$ simulations (where N is the number of different components) to determine the
contribution of each component to ice volume difference, therefore $2^2 = 4$ experiments are needed that systematically change
one variable. These experiments are listed in Table 2. The relative contributions of the initial conditions and climate can be
calculated by Eqs. (2) and (3):
$$dV_{ice} = \frac{1}{2}\big((V_i - V) + (V_{ci} - V_c)\big),\qquad (2)$$
$$dV_{climate} = \frac{1}{2}\big((V_c - V) + (V_{ci} - V_i)\big),\qquad (3)$$
To properly understand the effect of the initial conditions, we performed two sets of sensitivity experiments. In the first set,
labelled $V_{\_1}$, $V_{c\_1}$, $V_{i\_1}$ and $V_{ci\_1}$ (Table 2; experiments 3 – 6), both the topography and ice cover are set to be consistent between
the climate and ice sheet model components. Specifically, for the LGM, the Glimmer initial bedrock topography and ice surface
elevation was prescribed from the GLAC-1D reconstruction used in the FAMOUS LGM boundary condition. For the PGM,
the ice thickness data needed for the PMIP4 reconstruction to be converted to the Glimmer initial condition were not available.
Instead, both Glimmer and FAMOUS were initialised with the final timestep of the NROYa PGM (xpkyn) experiment since
it closely resembles the PMIP4 reconstruction. Experiment $V_{\_1}$ corresponds to a full LGM simulation and $V_{ci\_1}$ corresponds to
a full PGM simulation. In the second set of sensitivity experiments, we use the initial Glimmer ice sheet used in the ensembles,
i.e. the 18.2 ka mid-size ice sheet, only varying the FAMOUS initial ice sheets to see how this difference in orography between
the climate and ice sheet models may have impacted the result. These experiments are labelled $V_{\_2}$, $V_{c\_2}$, $V_{i\_2}$ and $V_{ci\_2}$ (Table
2; experiments 7 – 10), with $V_{\_2}$ corresponding to the LGM NROYa (xpken) and $V_{ci\_2}$ corresponding to the PGM NROYa
(xpkyn).

**3 Results**

**3.1 Ensembles**

Our ensembles of 62 North American Ice Sheet configurations spans uncertainty in model parameters and reveals the wide
range of possible modelled ice sheet evolutions. Over the full ensembles, we find that the set-up of the original Wave 2 meant
that the albedo values were too high and so the use of more realistic albedos in these ensembles led to many of the runs
deglaciating to very low volumes as shown in Fig. 5 (see Appendix D for more detail).

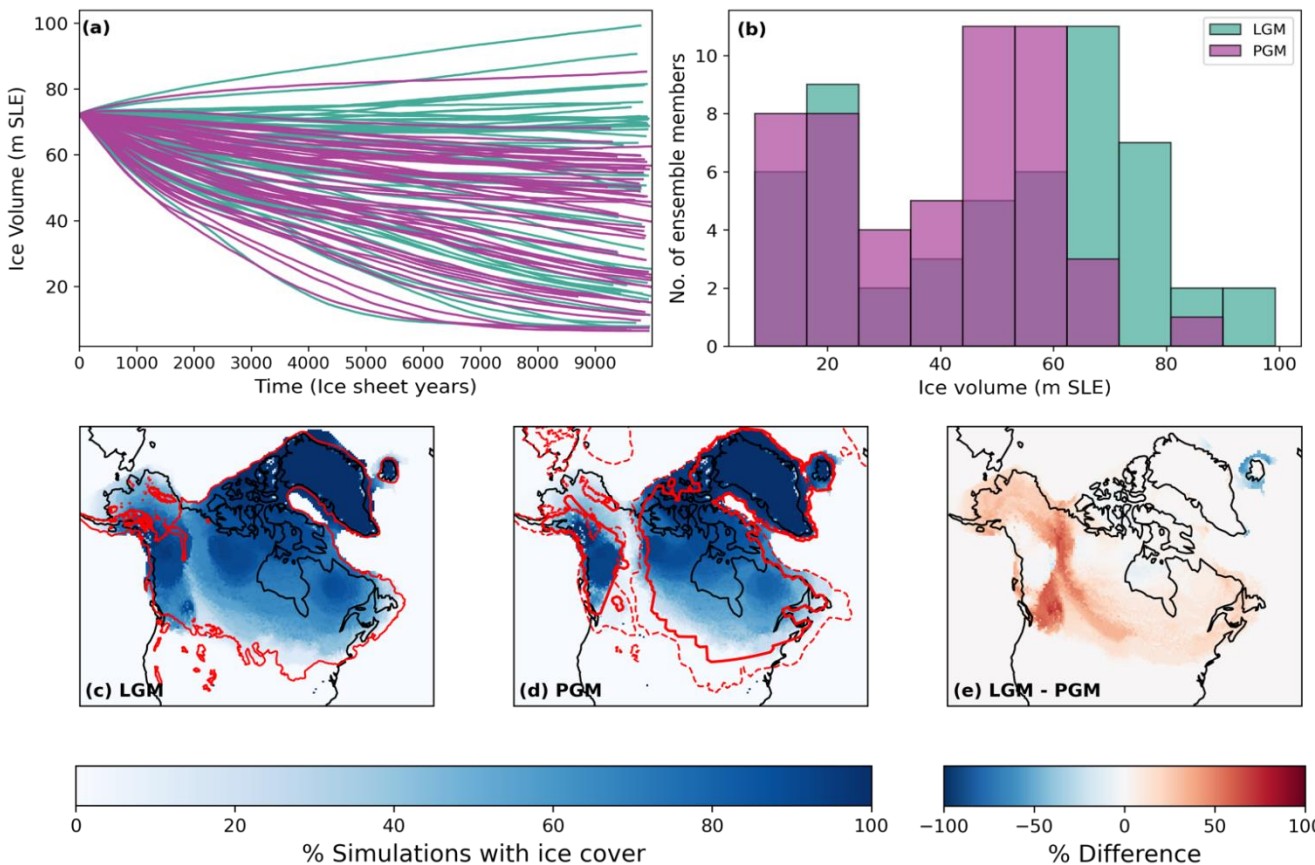


**Figure 5. (a) Ice volume evolution over modelled time, and (b) density distribution of final ice volumes for the full LGM and PGM ensembles. Percentage of simulations with ice cover for (c) LGM (with the Dalton et al., (2020) reconstructed margin shown in red); (d) PGM (with the PMIP4 PGM modelled margin shown in solid red and the Batchelor et al., (2019) reconstructed maximum MIS 6 margin shown in dashed red), and (e) the difference between the LGM and PGM, at the end of the simulations.**

**Table 4. Average volumes (NAIS + Greenland) and southern NAIS areas and their standard deviations (SD) of the NROY LGM and PGM simulations. Also shown are estimated values from literature for comparison.**

|  | *Mean Total Volume (SD), m SLE* | *Estimated Total Volume, m SLE* | *Mean Southern Area (SD), x 10⁶ km²* | *Estimated Southern Area, x 10⁶ km²* |
|---|---|---|---|---|
| *LGM* | 82.1 (8.29) | 61-98 (Rohling et al., 2017) | 5.55 (0.33) | 6.28 (Dalton et al., 2020) |
| *PGM* | 62.3 (10.3) | 49-69 (Rohling et al., 2017) | 3.64 (0.82) | 3.32 (Menviel et al., 2019) |

314

After applying our implausibility criteria (Sect. 2.4), six non-implausible or NROY LGM simulations remained. Table 4 gives the average volumes and areas of these six simulations and the corresponding six PGM ice sheets compared to estimated values from empirical and model data. All six LGM simulations show an overgrowth of ice in Alaska of varying magnitudes, as a result of the previously mentioned climate model bias. However, in other regions the simulations display a very similar ice extent, with the southern area only varying by 9.7 x $10^5$ km². None of the simulations form ice lobes, as expected, but they do show a close match to reconstructed ice extent in our target area, although towards the lower end of the plausible range, and in the marine regions (Fig. 6a and 7a). There is a minimum ice volume of 73.9 m SLE and a maximum of 97.1 m SLE. The maximum ice thickness varies by around 300 m but the overall shapes of the ice sheets remain the same, with the thickest ice towards the east of the ice sheet over Hudson Bay.

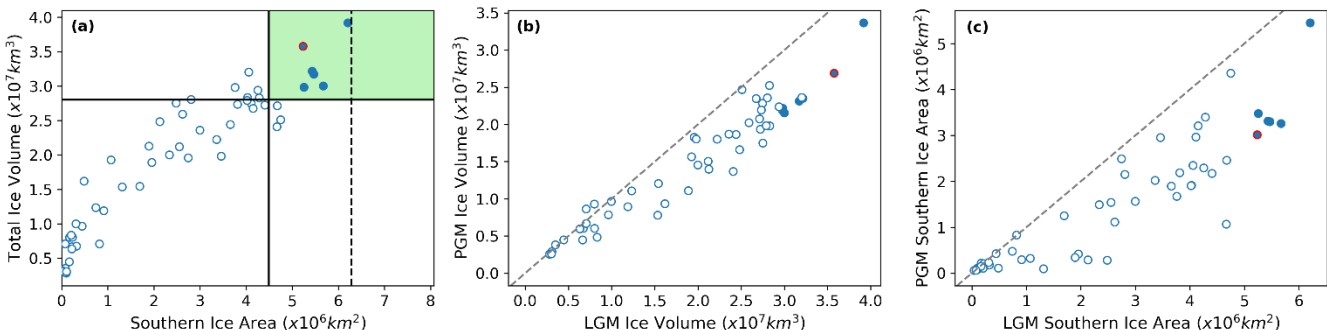

**Figure 6. (a) The relationship between final ice volume and southern area for the LGM ensemble, and the relationship between the LGM and PGM (b) final ice volume, and (c) final southern areas. The filled in blue dots represent the six NROY LGM simulations and the solid lines on panel (a) show the minimum volume and area constraints applied to the ensemble. The ensemble member chosen as NROYa is outlined in red (Sect 2.5).**

All the PGM ice sheets were smaller in volume than their LGM counterpart (Figs. 6 and 7) and displayed a smaller extent in the southern margin and the saddle region between the western Cordilleran Ice Sheet and eastern Laurentide Ice Sheet. However, the PGM simulations also displayed more variability in their ice extent and volumes. The ice volumes range from 53.4 m SLE to 83.37 m SLE and the southern extent varies by 2.44 x $10^6$ km². The range in maximum ice thickness is also over double the LGM, varying by around 613 m. These PGM configurations also look plausible compared to the less well constrained extent data available, including previous empirical and modelled reconstructions of the PGM/MIS 6 extent (Menviel et al., 2019; Batchelor et al., 2019; Fig. 7b). For example, all the simulations maintain an ice-free corridor between the Laurentide and Cordilleran ice sheets which is a common feature in these PGM reconstructions. In addition, the excess Alaskan ice seen in LGM simulations is also present at the PGM, however the growth is not as excessive.

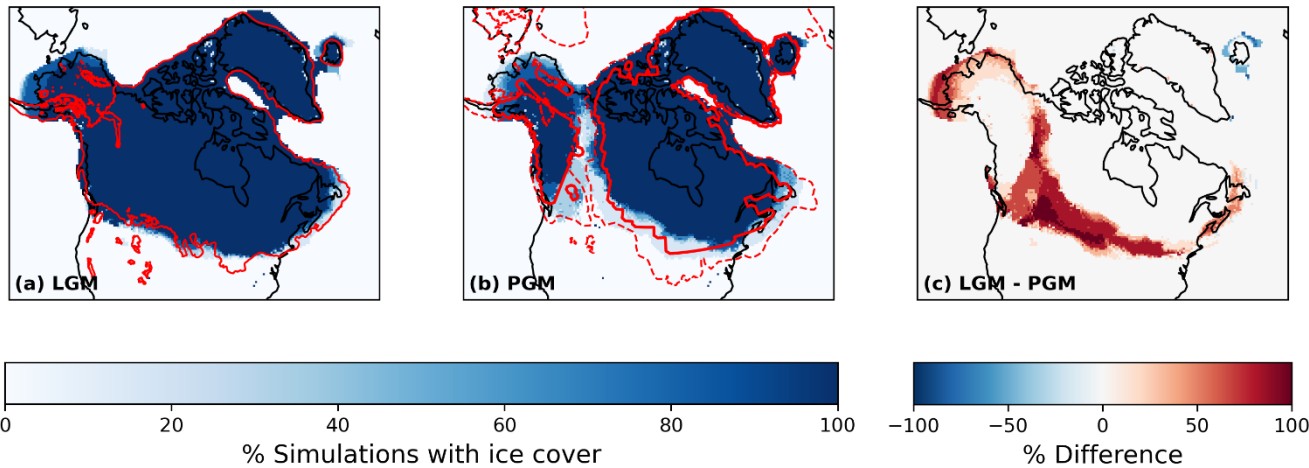

**Figure 7. Percentage of simulations with ice cover for (a) LGM with the Dalton et al., (2020) reconstructed margin shown in red; (b) PGM with the PMIP PGM modelled margin shown in solid red and the Batchelor et al., (2019) reconstructed maximum MIS 6 margin shown in dashed red, and (c) the difference between the LGM and PGM, at the end of the simulations for the six NROY ensemble members.**

### 3.2 Impact of initial ice sheet vs climate

Out of our six NROY model configurations, we selected the parameters of a pair of LGM and PGM experiments xpken/xpkyn (NROYa; Fig. 6) to perform two sets of four sensitivity experiments to decompose the effects of climate forcing and initial conditions on the final ice sheet volume. This included repeating xpken and xpkyn using matching FAMOUS and Glimmer LGM and PGM initial conditions respectively (Table 2, experiments 3 and 6). For both glacial maxima, using the matching initial conditions resulted in more excess ice over Alaska (Fig. C1), though the southern ice extents are relatively similar between the two sets of experiments. Overall, for the LGM, using the GLAC-1D reconstruction in Glimmer ($V_{-1}$) resulted in an ice sheet 9.7 m SLE larger than if the 18.2 ka ice sheet was used ($V_{-2}$) (Table 5; Fig. C1a). For the PGM, the matching initial conditions ($V_{ci\_1}$) resulted in only 0.45 m SLE increase from the NROYa simulation ($V_{ci\_2}$) due to a decrease in ice volume over the Laurentide ice sheet (Table 5; Fig. C1b).

**Table 5. Final ice volumes of the four sensitivity experiments performed with matching climate model and ice sheet model ice sheets and the equivalent four performed with different initial ice sheets in each model**

| Experiment | Final ice volume (m SLE) | Experiment | Final ice volume (m SLE) |
|---|---|---|---|
| $V_1$ (full LGM) | 100.3 | $V_2$ | 90.6 |
| $V_{c\_1}$ (LGM ice , PGM climate) | 104.2 | $V_{c\_2}$ | 97.1 |
| $V_{i\_1}$ (PGM ice, LGM climate) | 64.7 | $V_{i\_2}$ | 63.0 |
| $V_{ci\_1}$ (full PGM) | 68.6 | $V_{ci\_2}$ | 68.1 |

The final ice sheet volumes from the first set of four sensitivity experiments (Table 2; experiments 3 – 6) are displayed in Table 5 and shown in Fig. 8. The results of the second set of four experiments (Table 2; experiments 7 – 10) are also included in Table 5. The results of the factor decomposition analysis show that the simulated ice volume at the PGM was 31.7 m SLE ($1.25 \times 10^7$ km³) lower than at the LGM ($dV_1$). The initial ice sheet configuration ($dV_{i\_1}$) alone caused a 35% decrease in volume, but this was partially offset by the climatic conditions ($dV_{c\_1}$), which resulted in an increase in volume of 4%. The result was similar for the second set of experiments, with the initial ice sheet configuration ($dV_{i\_2}$) causing a decrease of 31% in ice volume at the PGM compared to the LGM, but the climate ($dV_{c\_2}$) caused a 6% increase in volume.

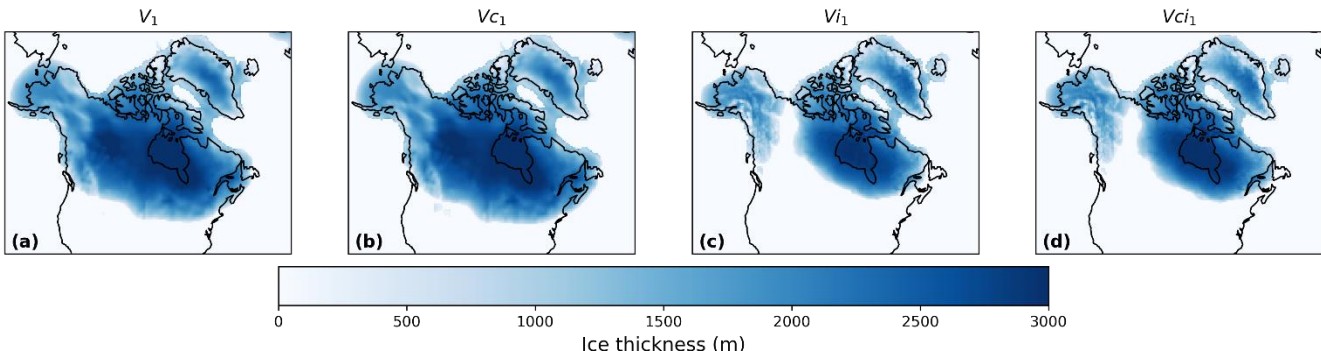

**Figure 8. Final ice thickness in the sensitivity tests using (a) LGM ice sheets and LGM climate; (b) LGM ice sheets and PGM climate; (c) PGM ice sheets and LGM climate, and (d) PGM ice sheets and PGM climate.**

The PGM climate is conducive to growing a larger ice sheet (Fig. 9a) because the orbital configuration results in the Northern Hemisphere receiving less incoming solar radiation in spring and early summer (Table 1; Fig 2a). This reduces the melting of snow that has accumulated in winter (Fig. 9b). The winter snow accumulation is also higher at the PGM than at the LGM (Fig. 9c) due to the PGM having warmer air temperatures in autumn and winter, because of the orbital forcing, leading to a wetter climate. Summer SSTs are also cooler at the PGM (Fig. 2c) due to lower spring insolation, further contributing to reduced

runoff. In contrast, the Greenland ice sheet decreases in size due to PGM climate conditions (Fig. 9a), likely due to higher sea
ice concentration south of Greenland reducing the moisture source available for precipitation.

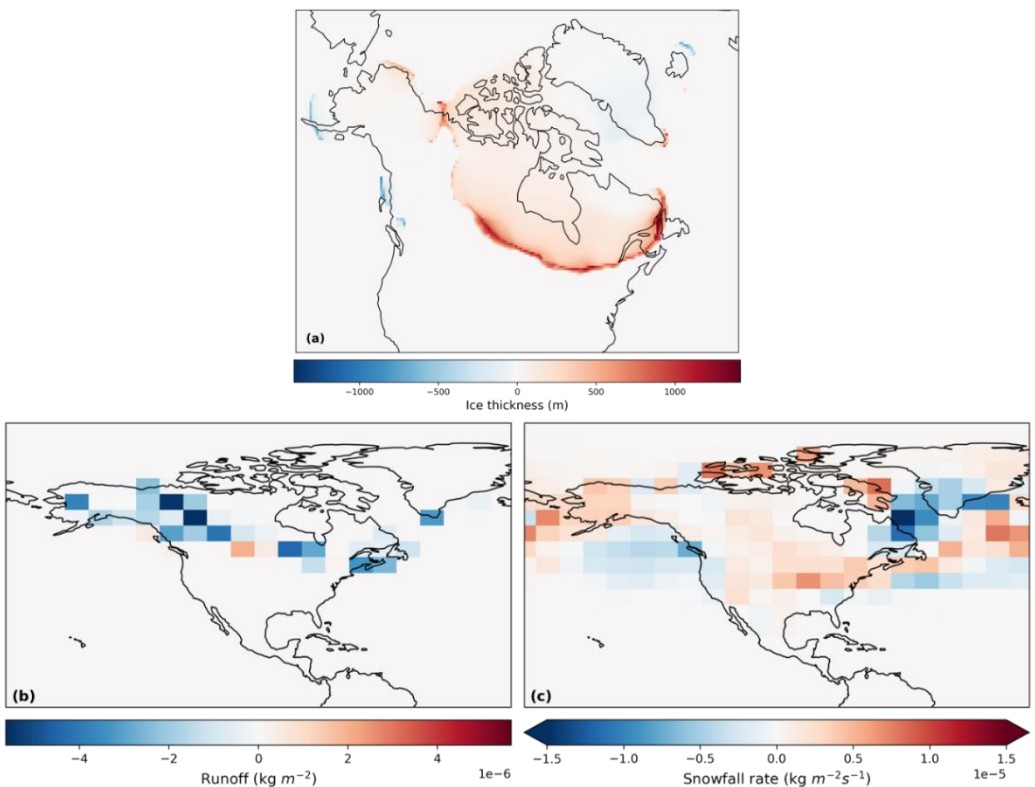


**Figure 9. Difference between experiment $V_{ci\_1}$ (full PGM) and $V_{i\_1}$ (PGM ice sheet with LGM climate) isolating the effect of LGM**
**climate vs PGM climate on (a) final ice thickness simulated by Glimmer and (b) spring (MAM) runoff and (c) winter (DJF) snowfall**
**over the first 10 years.**
**3.3 Uncertainty due to model parameters**
Due to the sampling strategy, this ensemble does not have an optimal design for analysing the sensitivity of the ice sheets
during the two time periods to the different model parameter values because our ensemble of simulations does not uniformly
span the uncertain parameter space. For this, we refer the reader to the studies of Gandy et al., (2023) and Sherriff-Tadano et
al., (2023), which present larger ensembles of experiments. Here, we first evaluate if our results are consistent with these two
studies before examining if the difference between the PGM and LGM ice sheets is sensitive to specific model parameters.
Based on correlations between the parameters and ice sheet area and volume, we find that the LGM and PGM behave similarly
across the parameter ranges (Figs. E1 and E2) and most of the uncertainty in the results for both periods can be explained by
parameters that affect the surface albedo of the ice sheet; *Daice, AV_GR* and to a lesser extent, *Fsnow*. Higher values of *Daice*
and *Fsnow* and lower values of *AV_GR* cause higher albedos and lead to larger ice sheets (Table 3). *Basal sliding* also
influences the volume of the ice sheet, with less impact on the area, with lower values and thus lower ice velocities causing
larger volume ice sheets. The cloud parameter *CW* also shows a relatively high positive correlation for the PGM (Fig. 10).
This is consistent with the findings of previous studies and current understanding on the importance of albedo for ice sheet
evolution (Willeit and Ganopolski, 2018; Sherriff-Tadano et al., 2023; Gandy et al., 2023).
Additionally, there is a negative correlation between the difference in ice volume and area between the LGM and PGM and
the parameters *AV_GR, basal sliding*, and *RHCrit*. Conversely, there is a positive correlation between the LGM-to-PGM
difference in ice volume/area and *Daice* (Fig. E3). This suggests that lower values of *AV_GR* and higher values of *Daice* and
thus a higher albedo, as well as lower ice sheet velocity and more cloud, make the ice sheet more sensitive to changes in
radiative forcings from the orbital boundary conditions.

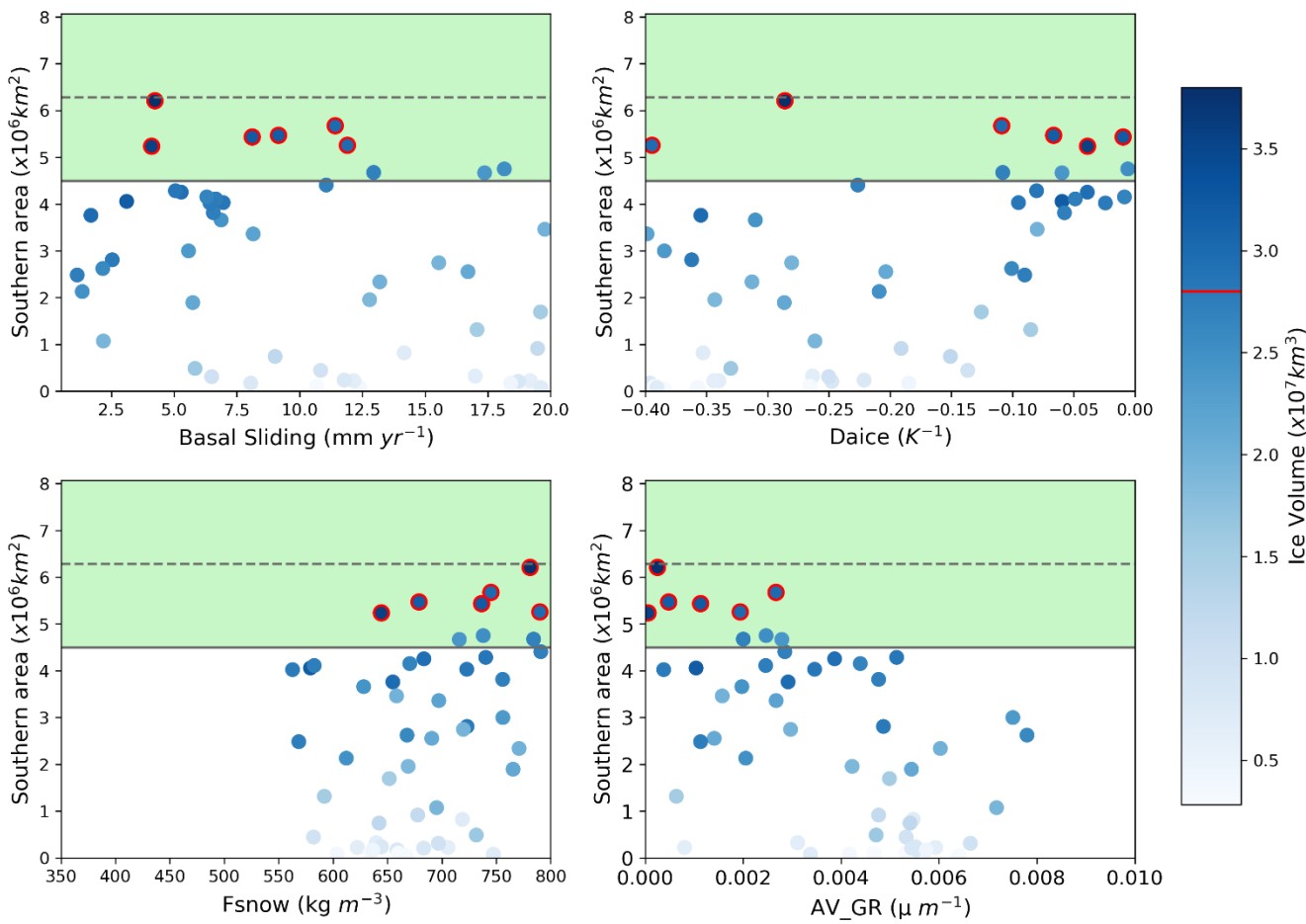


**Figure 10. Relationship between LGM southern area and the four most influential parameters. The green shaded region shows the**
**southern area constraint applied with the dotted line showing the exact area of the reconstruction and the solid line the minimum**
**bound applied. The colour scale represents ice volume and the dots outlined in red are the six NROY LGM simulations with the red**
**line on the colour bar showing the volume constraint.**

## 4 Discussion

After constraining our ensembles based on the available empirical and model data for the LGM, we find that the model was
able to successfully simulate the ice sheet at both periods under different LGM and PGM climate boundary conditions (orbital
parameters, SSTs and global orography) and initial ice sheets. However, the southern extents of the constrained LGM
simulations all fall towards the lower end of the plausible range, which is a common feature seen in other simulations using a
low resolution atmosphere model due to biases that cause a reduced stationary wave effect over this region (Ziemen et al.,
2014; Sherriff-Tadano et al., 2023; Gandy et al., 2023). Additionally, the ice lobes that are present over the Great Lakes are
not captured in these simulations. Again, this is common in ice sheet models and is likely a result of missing subglacial
processes or the low resolution of the climate and ice sheets models.
Analysis of the behaviour of the modelled ice sheets across the parameter spaces reveals that both the LGM and PGM ice
volume and extent have similar sensitivities to parameter uncertainties. We therefore conclude that parameters that produce a
good LGM NAIS also produce a plausible PGM NAIS under PGM boundary conditions and thus similar model parameters
are appropriate for use when modelling both periods. Our simulations can thus be compared and analysed to understand the
causes of the different configurations between the two periods. However, since the ice volume is most sensitive to surface
albedo and most simulations deglaciate under low values of *Daice*, this suggests that the value of bare ice albedo in the model
may need to be increased for future work.
The results of the sensitivity analysis show that the difference in initial ice sheet boundary conditions overwhelmingly
determined the difference in final ice volume between the LGM and PGM in the ensemble of simulations. We tested the impact
of starting from LGM and PGM ice sheet configurations in Glimmer instead of the 18.2 ka BP ice sheet and found that this
caused an even larger difference in ice volume between the two glacials. Comparing the simulations that use the same initial
ice topography in FAMOUS and Glimmer (first set of experiments), to those that use different topographies (second set of
experiments), whilst keeping the ice cover consistent, reveals that the relative contribution from the initial ice sheet boundary
conditions, compared to the climate conditions, to the simulated differences between the LGM and PGM ice sheets, remains
similar. This suggests that the dominant feedback responsible for this result is the ice-albedo feedback rather than the
temperature-elevation feedback. A similar conclusion was obtained by Abe-Ouchi et al., (2007) who studied the relative
contribution to climate over ice sheets from the ice sheet itself and the orbital parameters and $CO_2$ concentration. They found
the cooling caused by the ice sheet themselves was the dominant effect, mostly due to albedo feedbacks, which increase with
ice sheet area. Kageyama et al., (2004) also highlighted in their study the importance of the albedo feedback on the maximum
modelled North American ice volume. They show that changes in vegetation are needed to initiate glaciation over North
America which is then accelerated by the ice-albedo feedback. The North American ice sheet was larger at the LGM than at
the PGM. However, this sensitivity analysis reveals that the difference in orbital parameters, GHGs and SSTs (climate)

between the LGM and PGM encourages the growth of a larger North American ice sheet at the PGM (Fig. 9a). This effect would likely be even stronger if we had used the orbit at 137 ka BP (the timing of the minimum in Northern Hemisphere summer insolation; Fig. 11a-c) since the PGM would have received even lower insolation in spring and early summer. This result highlights the importance of the evolution of these climate factors and the ice sheets during the preceding glacial cycles in determining the glacial maxima configurations. For example, during the start of the Last Glacial Cycle (MIS 5; ~115-80 ka BP), the variation in 65° N summer insolation was relatively large as a result of changes in orbital parameters (Fig 11a-c), which resulted in multiple cycles of growth and recession of the North American Ice Sheets during this period, but total ice volume remained low (Bonelli et al., 2009; Ganopolski et al., 2010; Dalton et al., 2022). Insolation then reaches a minimum at ~70 ka BP (Fig 11c) which, combined with decreasing concentrations of $CO_2$ (~190 ppm at ~65 ka BP; Fig. 11f), led to a significant increase in ice sheet volume to almost LGM extent (Fig. 11d) and a switch to more widespread glacial conditions at the MIS 5/MIS 4 transition (Bonelli et al., 2009; Dalton et al., 2022). The size of the NAIS at this time was large enough to induce positive feedbacks, such as the ice-albedo feedback, allowing its maintenance throughout MIS 4 and MIS 3 (~70-30 ka BP) despite an increase in insolation from ~50-30 ka BP (Fig. 11c). This was also supported by a continued decrease in $CO_2$ (Fig. 11f). Growth of the ice sheet could then continue to its glacial maximum extent following a further insolation and $CO_2$ decrease during MIS 2 (~30-21 ka BP) (Fig. 11c-f). In contrast, prior to the PGM there were peaks in insolation at ~172 and ~148 ka BP that reached higher levels than were reached prior to the LGM during MIS 4 and MIS 3 (Fig. 11c; Berger; 1978). This may have inhibited an initial significant build-up of ice over North America, as during MIS 4, preventing the initiation of an ice-albedo feedback strong enough to enable the continued growth towards a larger LGM configuration and/or maintain its volume through the second insolation peak. In addition, there was more time between the LGM and the insolation maximum at ~50-30 ka BP compared to the PGM and the maximum at ~147 ka BP. Therefore, the PGM NAIS may have not had enough time to regrow before insolation started to increase again. Thus, investigation of the processes and interactions that took place prior to the glacial maxima will be needed to fully understand why the LGM and PGM NAIS configuration differed.

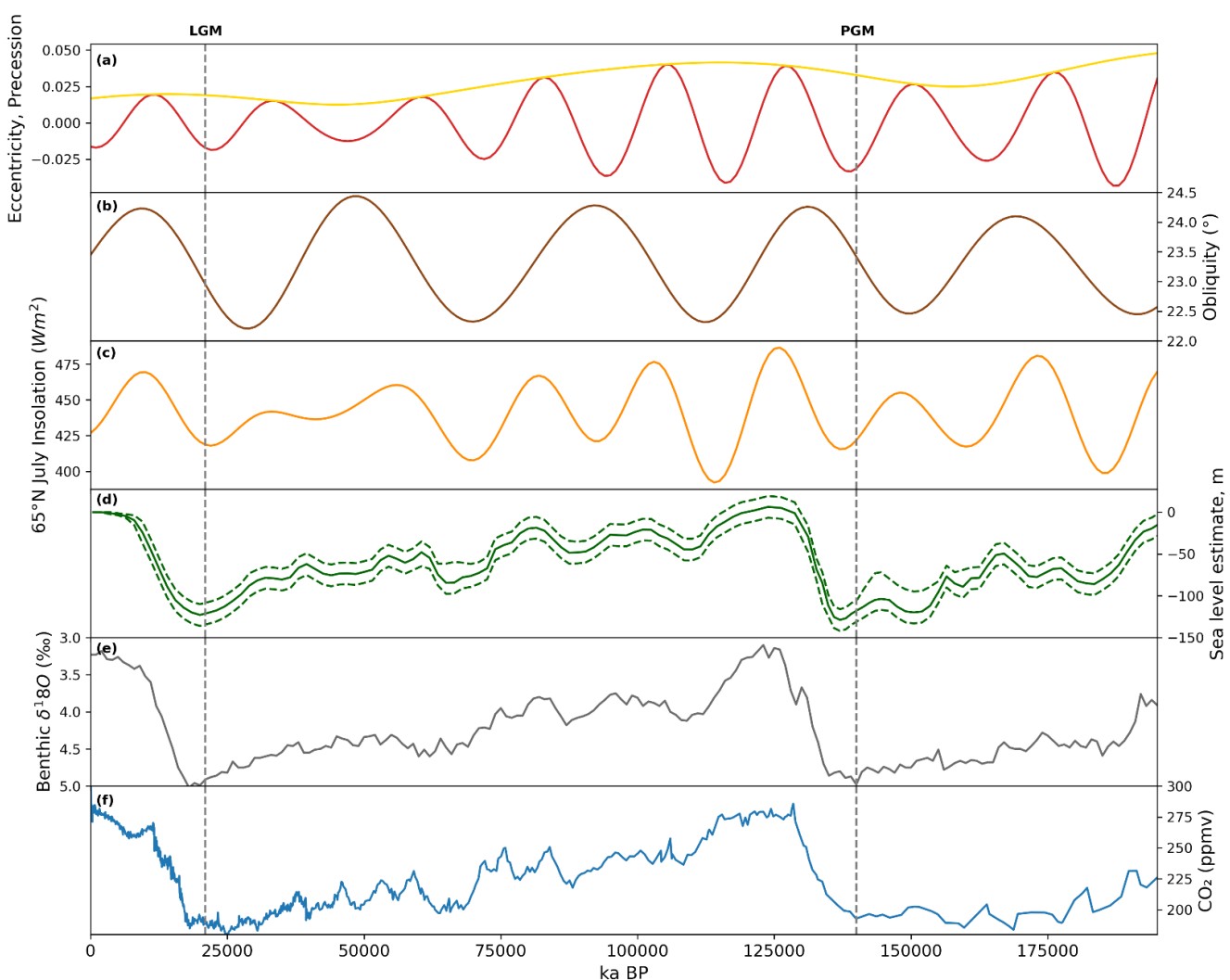

457

**Figure 11. Evolution of climate proxies over the last two glacial-interglacial cycles: (a) precession index (red) with eccentricity as an envelope (yellow); (b) obliquity (Berger, 1978); (c) July insolation at 65° N (Berger and Loutre, 1999); (d) reconstruction of global mean sea level and uncertainty estimate (dotted lines) (Waelbroeck et al., 2002); (e) benthic δ18O global stack record (Lisiecki and Raymo, 2005), and (f) EPICA Dome C carbon dioxide ice core records (Luthi et al., 2008; Bereiter et al., 2015). The PGM and LGM are indicated by the dotted line.**

Additional feedbacks that played a role in the development of glacials into either an LGM-like or PGM-like mode are also missing in these simulations due to computational constraints. For example, the low resolution of the atmospheric component of FAMOUS means that it is capable of performing ensembles and long palaeo runs while directly coupled to an ice sheet model. However, it also means that many small-scale atmospheric processes (e.g. stationary wave response) caused by and affecting the ice sheet topography are not represented well (Kageyama and Valdes, 2000; Liakka and Nilsson, 2010; Beghin

et al., 2014; 2015; Liakka et al., 2012; 2016). Additionally, the shallow ice approximation used in Glimmer means that the ice sheet will not be able to simulate marine instabilities of advance and retreat (Pattyn et al., 2012). This effect will be minimal for the NAIS, but a more advanced ice sheet model would be required to simulate a marine ice sheet like the EIS.

As a reminder, the vegetation was kept fixed at pre-industrial distributions, but the vegetation prior to and next to the ice cover has been shown to be very important for determining ice sheet expansion in models through the vegetation-albedo feedback (Kageyama et al., 2004; Colleoni et al., 2009b; Horton et al., 2010; Stone and Lunt, 2013). Therefore, implementing glacial maxima distributions or dynamical vegetation may affect the results since the reduction in forest and expansion of tundra/shrubs compared to present day would increase the albedo of the surface next to the ice and affect the climate (Meissner et al., 2003). Similarly, the prescribed SSTs and sea ice concentrations used introduce an additional source of uncertainty. As well as impacting the global mean temperature and precipitation patterns in the simulations, the SSTs and sea ice used can have local climate impacts that affect the simulated ice sheets. This includes causing a warming or cooling over the more coastal areas affecting the melt rate, and impacting evaporation rates, which affects the amount of snowfall the ice sheets receive. The SSTs used in this study are cooler (as a global average) than the multi-proxy and data assimilation LGM SST reconstructions of Tierney et al., (2020) and Paul et al., (2020) and the constrained statistical reconstruction of Gandy et al., (2023) and Astfalck et al., (2024). HadCM3 also tends to simulate cooler SSTs compared to other PMIP4 models, although they are similar to CESM1.2 (Kageyama et al., 2021). Therefore, the use of colder SSTs in this study causes lower global mean temperature overall, but also would have caused a cooling next to the ice sheets and reduced snowfall, which would have impacted the ice sheet growth in different ways (Marsiat and Valdes, 2001; Hofer et al., 2012; Astfalck et al., 2024). The latter impact was shown to be most dominant in the study by Astfalck et al., 2024, suggesting that our simulated ice sheet volumes may have been larger had we used their warmer LGM SST reconstruction, due to increased evaporation. Prescribing the ocean forcing also neglects any effects changes in ocean conditions and ice sheets have on each other (e.g. Timmerman et al., 2010; Colleoni et al., 2011; Ullman et al., 2014; Sherriff-Tadano et al., 2018; 2021). Using a dynamical ocean would include the effects of meltwater and changes in atmospheric circulation, arising from the ice sheets, on ocean circulation and temperature, which would in turn affect the climate, feeding back onto the ice sheets themselves. Further work will be required to investigate the feedbacks between ice sheets and sea surface at the PGM, but this is beyond the scope of this study. We recommend the use of a fully coupled atmosphere-ocean-vegetation-ice sheet model to further investigate these feedbacks. The effect of dust deposition and ice dammed lakes have also been shown to have a large influence on the build-up of ice (e.g. Krinner et al., 2004; 2006; Naafs et al., 2012; Colleoni et al., 2009a) however further model developments would be needed to investigate these effects.

Finally, the Eurasian ice sheet also displayed important differences between the LGM and PGM and had a large influence on the climate. It is likely that some of the differences in the configurations of the NAIS and EIS between the two glacial maxima resulted from their interactions with each other (Beghin et al., 2014; 2015; Liakka et al., 2016). To investigate the EIS at the PGM, we recommend the use of an efficient marine ice sheet model such as BISICLES that uses Adaptive Mesh Refinement

to refine the processes occurring at marine margins that are more important for the marine based Eurasian ice sheet (Cornford et al., 2013; Gandy et al., 2019).

**5 Conclusions**

We have performed and compared ensemble simulations of the LGM and PGM using a coupled atmosphere-ice sheet model (FAMOUS-ice) with prescribed surface ocean conditions and interactive North American and Greenland Ice Sheets. We tested the relative importance of the initial ice sheet configuration versus the climate boundary conditions on the resulting ice sheet volumes through sensitivity tests and factor decomposition analysis. The main conclusions of this study are as follows:

1. Successful simulations of the LGM and PGM North American and Greenland ice sheets are produced using a coupled climate-ice sheet model. We find that uncertain model parameters tuned to produce a plausible LGM North American Ice Sheet also perform well for the PGM.

2. The initial ice extents used as boundary conditions in coupled climate-ice sheet simulations have a much larger impact on the modelled NAIS than the climate boundary conditions, causing a ~30% decrease in ice volume at the PGM compared to the LGM. This is due to the ice-albedo feedback.

3. The climate of the PGM causes an increase in NAIS ice volume of ~6% compared to the LGM due to the orbital configuration causing the Northern Hemisphere to receive less insolation in spring and early summer. Since the LGM ice sheet was larger than the PGM, this suggests that the climate and ice sheet evolution prior to the glacial maxima contributes to the differences seen between the LGM and PGM ice sheets.

**Appendix A: Eccentricity equation correction**

The equation for the role of eccentricity on solar insolation used in the simulations in this paper was:

$$S(t) = S_o \left( \left( 1 + \frac{e^2}{2} \right) (1 + e \cos v) / (1 - e^2) \right)^2 \tag{4}$$

However, this is incorrect and has now been corrected in the model to:

$$S(t) = S_o \left( (1 + e \cos v) / (1 - e^2) \right)^2 \tag{5}$$

where; S(t) is the incoming solar insolation, $S_o$ is the solar constant, e is the eccentricity of the earth's orbit and *v* is the true anomaly (the angle of earth's current position on its orbit).

The PGM experiment 'xpky0' was re-run with the correct equation and shows that on average the SMB was slightly lower in our simulations than it should have been (decreased by 16% at the end of the simulations), leading to slightly smaller ice sheets

(Fig. A1). However, the impact is small (and would be even smaller for the LGM given the lower eccentricity) and does not
affect our overall conclusions.

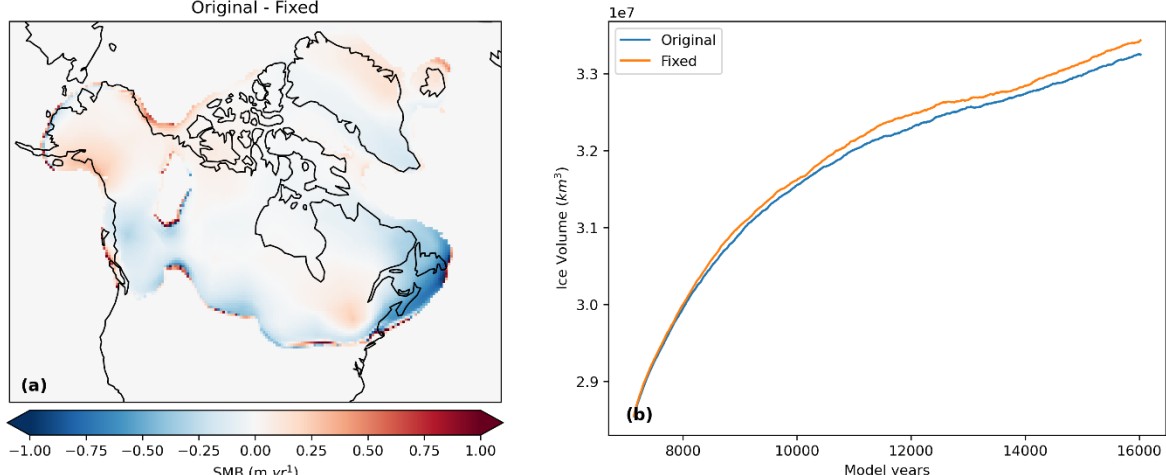

**Figure A1. (a) Difference between the SMB at the end of the experiments between the original simulation and the simulation using**
**the corrected eccentricity equation and (b) the evolution of ice sheet volume for both experiments.**
**Appendix B: Sea surface temperatures**

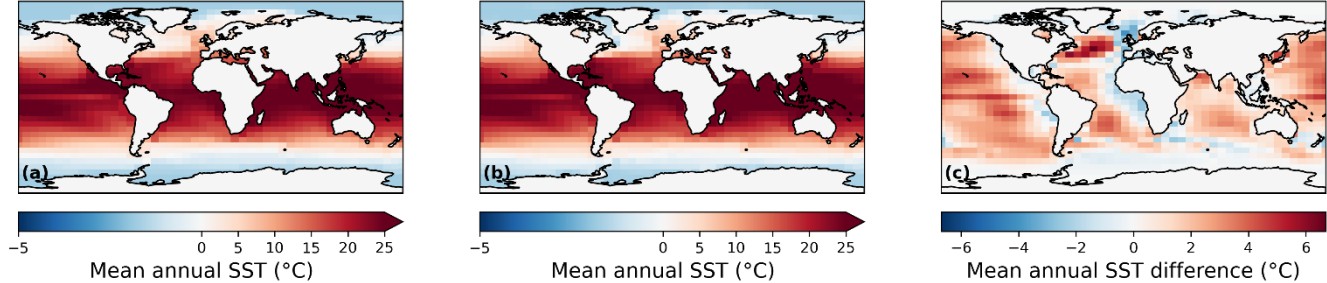

**Figure B1. Mean annual SSTs used in this study from HadCM3 for (a) LGM and (b) PGM and (c) the difference between the LGM**
**SST reconstruction used in Gandy et al., (2023) and the HadCM3 LGM SSTs.**

## Appendix C: Impact of different initial ice sheets

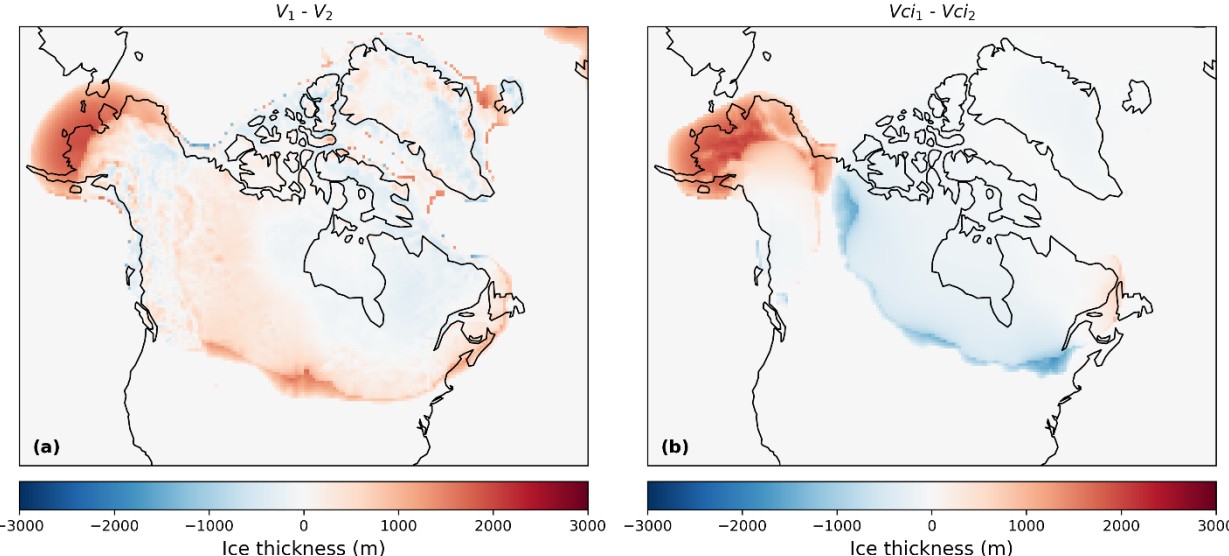

**Figure C1. Difference in the final ice thickness between the simulations with matching initial conditions in FAMOUS and Glimmer and the NROYa ensemble member for (a) the LGM and (b) the PGM.**

## Appendix D: Wave 2 methodology

The ensemble design in this study was based on the 'Not Ruled Out Yet' (NROY) parameter combinations from a second wave of ensemble members that followed on from the 280 member ensemble performed in Gandy et al., (2023). From the first wave of simulations, only 18 out of these 280 members produced a large enough LGM North American Ice sheet to meet the volume and extent criteria they imposed (see details in reference). Further work was thus performed to augment the ensemble of simulations that met the NROY criteria. We used statistical emulation to identify plausible regions in the parameter space. As there was limited information to constrain the domain of plausibility in the parameter space, we instead implemented an early-stopping criteria that allowed us to prevent the full execution of model runs that were not expected to produce good ice sheets. To do this we first modelled, from Wave 1, the predicted equilibrium area of the ice sheet from the value of the initial surface mass balance. Mathematically, we specified;

$$A = f(b) + \epsilon, \tag{6}$$

where A is the 'equilibrium' ice sheet area after 10,000 ice sheet years, b is the 20 year averaged SMB value over the ice sheet and $f(\cdot)$ may be any function. We considered $f$ to be either linear or sampled from a Gaussian Process (GP) and found the linear model gave more conservative uncertainty estimates which was desired since the Wave 2 runs needed to bound the

NROY space. The predictive interval for the model is $P(b) = [f(b) + 3\sqrt{\text{var}(\epsilon)}, f(b) - 3\sqrt{\text{var}(\epsilon)}]$ and we targeted equilibrium ice sheet areas in the interval $T = [1.5 \times 10^7 \text{ km}^2, 2 \times 10^7 \text{ km}^2]$. The interval $T$ is analogous to the target interval defined using Pukelsheim's 3-sigma rule in standard history matching (Pukelsheim, 1994). Plausible values of $b$ satisfy the condition that $P(b) \cap T$ is non-zero, that is, for $b$ to be plausible, the predictive bound $P(b)$ and the plausible equilibrium ice sheet area $T$ must intersect. It was found that the 20-year averaged SMB had to be at least positive to produce a plausible ice sheet.

To further improve efficiency, we used statistical emulation to produce plausible values of $b$ (and hence equilibrium ice sheet areas); iterating the training data of the emulator with each wave of simulator runs. Define by $x$ the multivariate vector of parameters that they build the emulator over: here $x$ comprised of the 4 most influential parameters *Fsnow, AV_GR, Daice,* and *Flow Factor*. We model $b$ with a random error process, $b \sim GP(x) + \eta$, where the effects of the parameters not explicitly represented in $x$ are handled by the stochasticity of the process represented by $\eta$. Values of $b$ were sampled using a stratified k-extended Latin Hypercube design (Williamson, 2015) and three sub-waves were executed, from which, a candidate set for the Wave 2 ensemble was extracted.

The first sub-wave (Wave 1.1) samples 200 ensemble members, which are predicted from the emulator to have non-negligible probability of positive SMB. This results in around 50% of simulations in this sub-wave having a positive SMB, an increase from 15% in the original wave (Fig. D1, Wave 1.1). We attempt to refine the predictive bounds on the GP model twice more (Fig. D1, Wave 1.2 and 1.3), with no improvement. This is likely due to the inherent stochasticity of the climate model and cumulative effects of the parameters that they absorb into the predictive error term. At the end of this process of iterative short waves, the candidate set contains over 1000 20-year long simulations that have a positive SMB over the North American ice sheet. From this candidate set, and again using stratified k-extended Latin Hypercubes, we select an optimal (with respect to space-filling and accounting for the previous Wave 1 runs) design of 200 ensemble members to continue for a full 10,000 years to an equilibrium North American Ice Sheet. These 200 simulations make up the Wave 2. For context, this workflow of GP model sub-waves saved around 230,000 core hours (or about two months of real time) compared to running a full second ensemble wave.

Out of these 200 Wave 2 simulations, 176 members were identified to be NROY based on the original volume and extent thresholds. It is based on these results that we sub-sampled 62 parameter combinations for our simulations. This number of simulations was selected to enable us to run long equilibrium LGM and PGM simulations over a full ensemble within reasonable computational requirements. From the 176 NROY parameter combinations we randomly generated $10^7$ candidate designs of size 62 from which we selected an approximate maximin design. This is obtained by: first linearly transforming each parameter onto the same range of [0, 1] to aid comparability; before computing the minimum distance between a parameter vector and its nearest neighbour; and then selecting the candidate design that maximised this distance. The resulting design possesses parameter vectors which are well-spaced and thus adequately cover the NROY space.

Our simulations use slightly different orbital parameter values and sea surface conditions to that of Gandy et al., (2023) (see Sect. 2.3). Thus, we do not expect the sample of 62 parameter combinations to provide full coverage of the NROY space but,

as seen in Sect. S2 of the supplementary information in Gandy et al., (2023), the output trends are sufficiently similar that we
expect this to be close enough to an optimal sample. Whilst we may have also sampled some parameter combinations outside
of the NROY space, we feel these will still provide valuable information about uncertainty in outputs at the LGM and PGM.
Our detailed comparison to empirical evidence and other model data (see Sect. 2.4 and 3.1) identified six parameter
combinations that match our criteria for LGM and PGM ice extent and volume, thus demonstrating the success of this approach.
Further exploration of the parameter space may produce NROY simulations in a different part of the parameter space but
would not change the conclusion of this paper.
Upon analysing the results, we found a technical error in the original Wave 2 ensemble which resulted in the values of the
parameter *Daice* being shifted from its intended range of –0.4-0 K$^{-1}$ to 0-0.4 K$^{-1}$, this means that the albedo of the bare ice was
increasing with melting, which is likely not the case. This produced larger values of surface albedo and thus larger ice sheets
in these Wave 2 simulations (not shown here). In the ensemble of simulations presented here, we corrected the *Daice* values
to match the intended parameter range. In some simulations, the switch of *Daice* value from a large positive number to a large
negative number would have resulted in a decrease in surface albedo and resulting ice sheet volume. This effect is negligible
for values of *Daice* closer to zero.

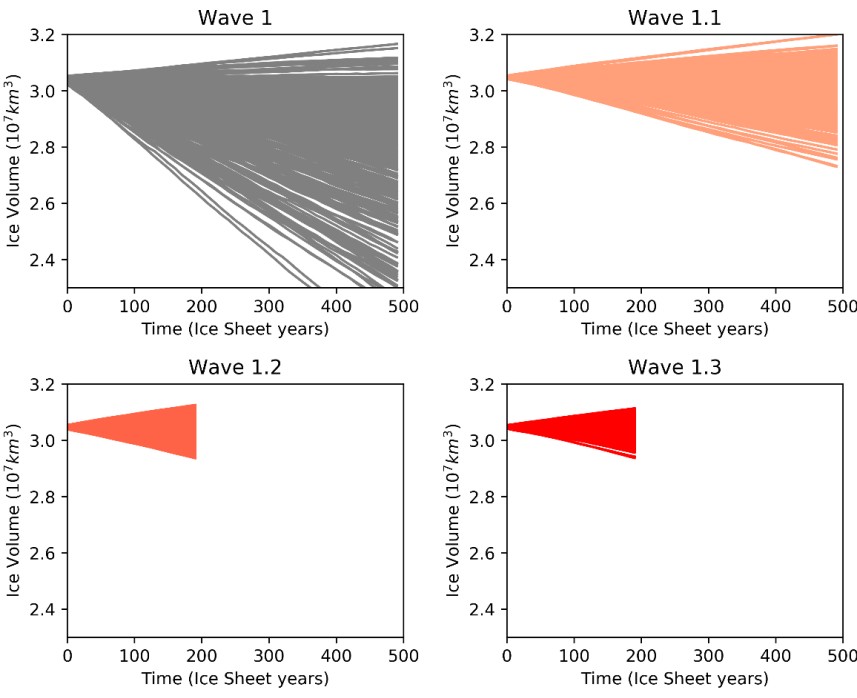


**Figure D1. Ice volumes simulated in the successive ensemble sub-waves of simulations sampled to have a positive initial surface mass**
**balance using the Gaussian Process emulator**

## Appendix E: Metrics vs parameters plots

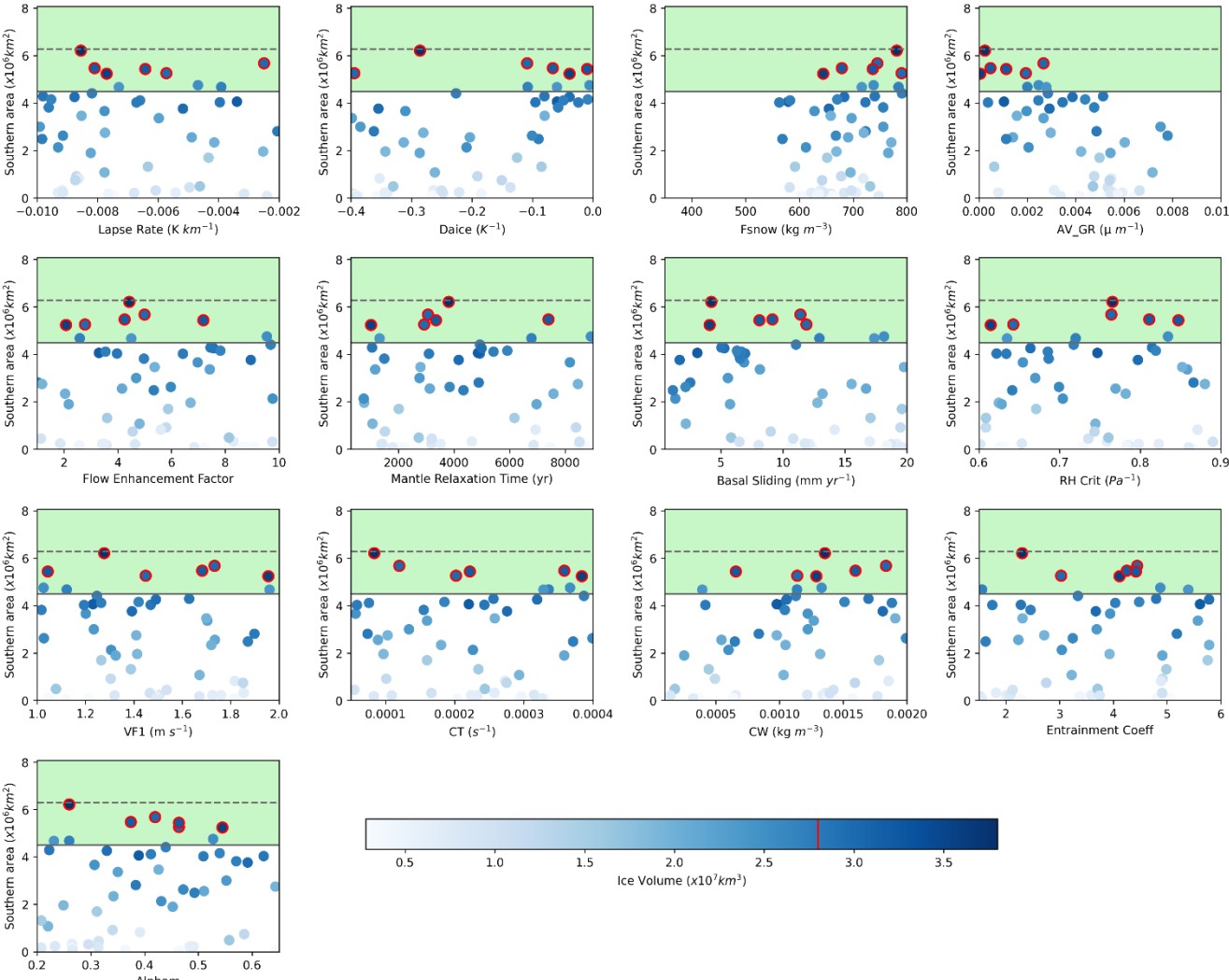

**Figure E1. Southern area versus each of the 13 parameters varied for the LGM ensemble. The green shaded region shows the southern area constraint applied with the dotted line showing the exact area of the reconstruction and the solid line the solid line the minimum bound applied. The colour scale represents ice volume and the dots outlined in red are the six NROY LGM simulations with the red line on the colour bar showing the volume constraint.**

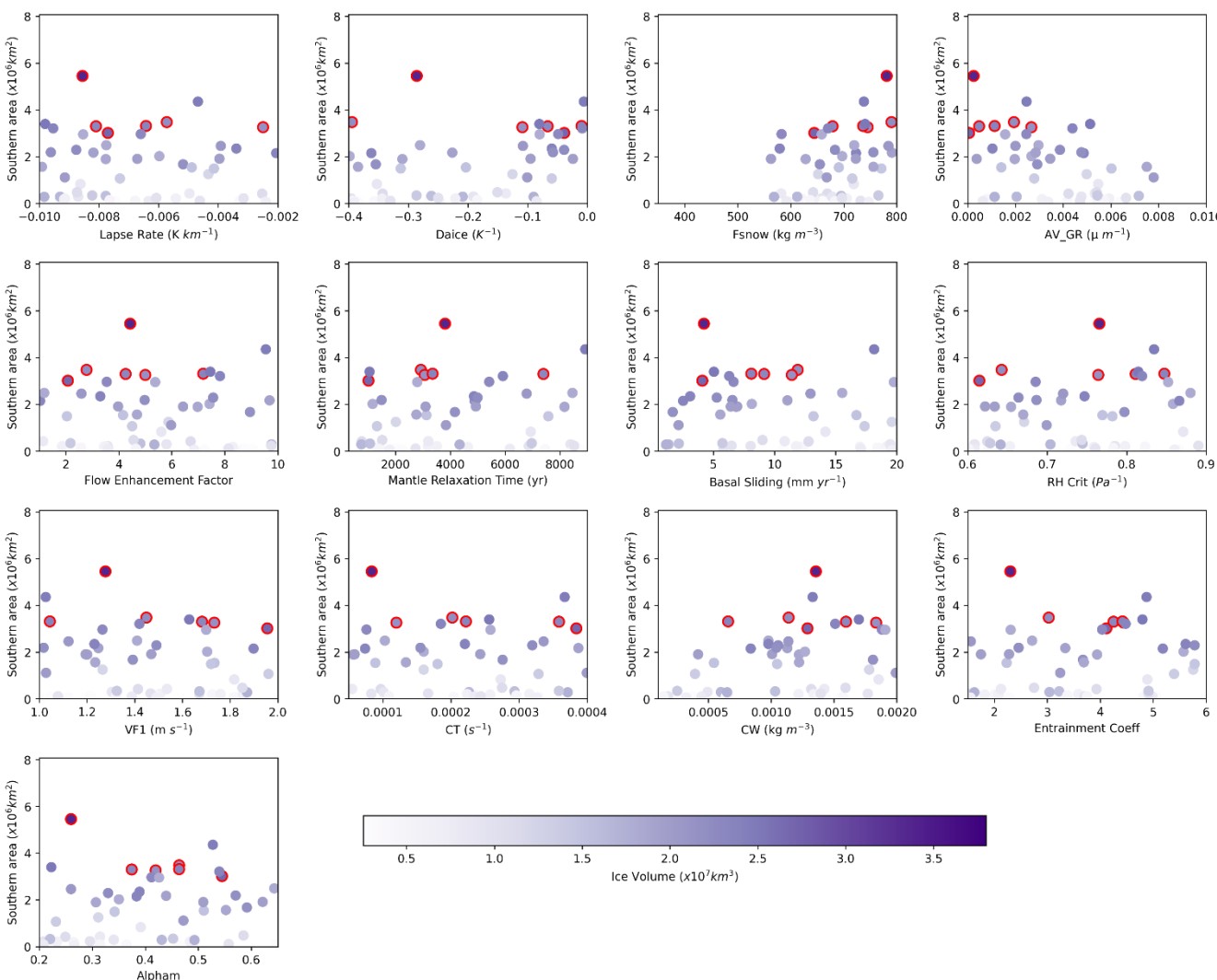


**Figure E2. Southern area versus each of the 13 parameters varied for the PGM ensemble. The colour scale represents ice volume**
**and the dots outlined in red are the corresponding six NROY PGM simulations.**

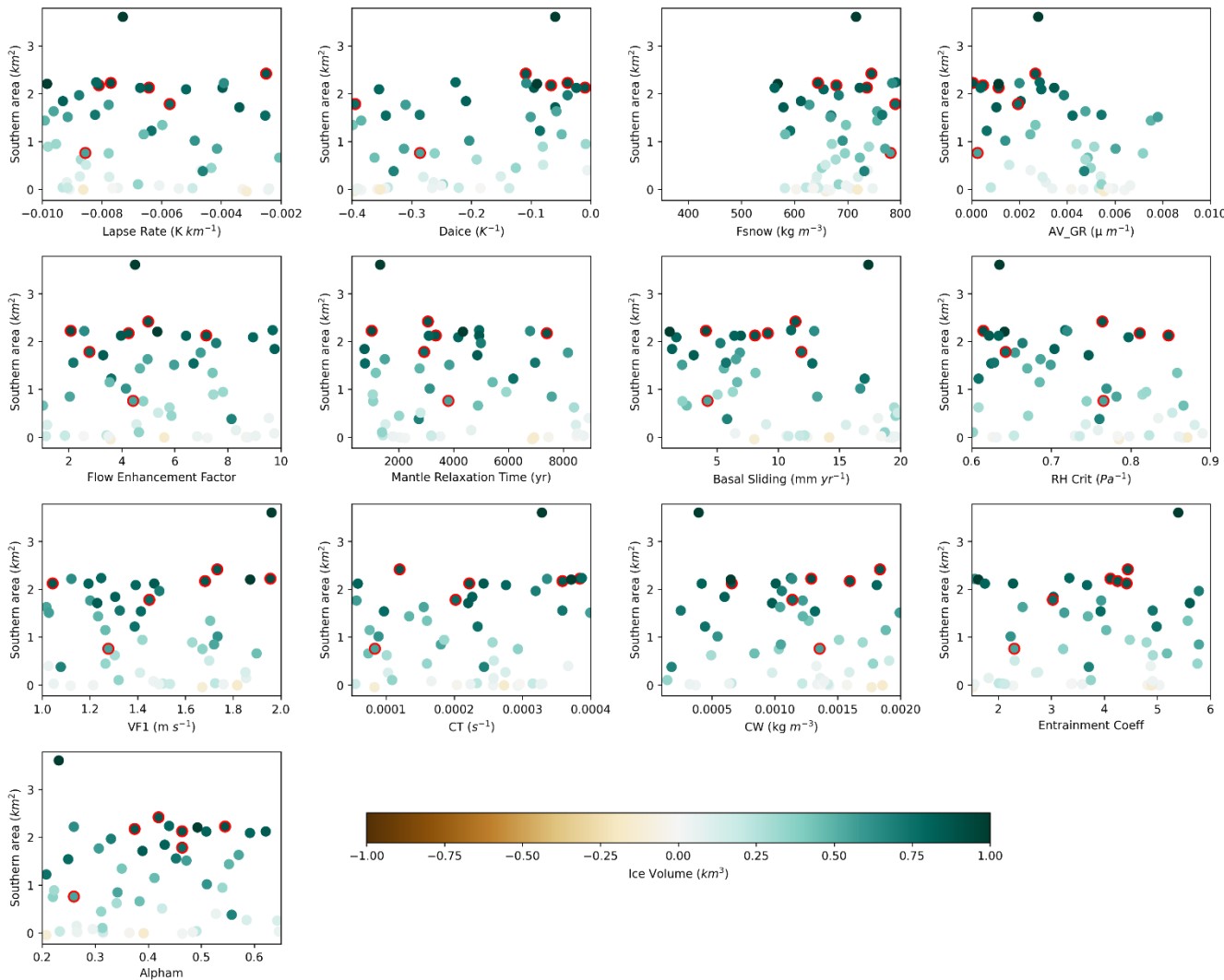


**Figure E3. Difference in southern area versus each of the 13 parameters varied between the LGM and PGM ensemble members.**

**The colour scale represents difference in ice volume and the dots outlined in red are the six NROY simulations.**

## Data availability

The boundary conditions used in this study as well as the full ensemble ice sheet model output and volume and extent metrics, climate timeseries for the NROY simulations and final ice volume data from the sensitivity tests are available at doi:10.5285/5e48b31e413b480792e4156191b654f4. All other model output data are available on request.

## Author contributions

VLP lead the project and performed the majority of the work. VLP, LJG, RFI and NG designed the simulations and VLP and NG prepared the initial and boundary conditions with support from OGP. VLP ran the simulations and analysed the results. LCA and NG designed and performed the Wave 2 simulations the ensembles were sampled from, and JO did the sampling. RSS provided technical and scientific support and updates for FAMOUS-Glimmer. PJV provided the PGM HadCM3 sea surface temperature and sea ice dataset. VLP wrote the manuscript with comments and contributions from all co-authors. LJG, RFI and NG supervised the project and LJG acquired the funding.

## Competing interests

The authors declare they have no conflict of interest.

## Acknowledgements

This research is primarily funded by the 'SMB-Gen' UKRI Future Leaders Fellowship MR/S016961/1, with LJG, JO, NG and LCA supported by the award and VLP's PhD studentship funded by the University of Leeds. RFI and RSS's contributions were supported by the RISICMAP19 NERC standard grant NE/T007443/1. LCA is supported by the ARC ITRH for Transforming energy Infrastructure through Digital Engineering (TIDE), Grant No. IH200100009. OGP is funded by the European Union's Horizon 2020 research and innovation programme RiSeR project (grant agreement no. 802281). The simulations were run on the high-performance research computing facilities of the University of Leeds and technical support was provided by Richard Rigby from the Centre for Environmental Modelling and Computation (CEMAC). The authors would like to thank Michel Crucifix, Peter Hopcroft, Pam Vervoort and Paul Valdes for discovering the eccentricity equation error and providing the corrected equation.

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
