# Peer review of "Contrasting the Penultimate and Last Glacial Maxima (140 and 21 ka"

_Climate of the Past, 2024_

## Author Comment (AC1)

**Response to reviews of 'Patterson et al., Contrasting the Penultimate and Last Glacial Maxima (140 and 21 ka BP) using coupled climate-ice sheet modelling'**

We thank both reviewers for their constructive and helpful comments, which we have used to improve our manuscript. We were happy to hear that the reviewers think our use of coupled climate-ice sheet modelling is an "improvement on existing literature" and "of great value and informative for the community". Following the reviewers' suggestions, we propose the following main changes:

1. Better description of our coupling procedure and clearer justification of the choices of initial and boundary conditions.
2. Adjusting the manuscript's outline to make the results from our factorial decomposition (original section 3.4 Sensitivity Analysis) the central feature of our manuscript. This includes moving some text from the results sections to the methods (new sections 2.4 and 2.5), moving old section "3.3 Climate-ice sheet interactions" to the methods and appendix and moving the section "Uncertainty due to model parameters" after the main results section "3.2 Impact of initial ice sheet vs climate". The new structure of the manuscript will be as follows:

**1 Introduction**

**2 Methods**
    2.1 Model description
    2.2 Experiment design
        2.2.1 Climate boundary conditions
        2.2.2 Ice sheet boundary conditions *(New section)*
    2.3 Ensemble design
    2.4 Implausibility metric *(methods text moved from section 3.1.2)*
    2.5 Sensitivity analysis *(methods text moved from section 3.4)*

**3 Results**
    3.1 Ensembles
        3.1.1 Unconstrained ensembles
        3.1.2 Constrained ensembles
    3.2 Impact of initial ice sheet vs climate *(previously results from section 3.4)*
    3.3 Uncertainty due to model parameters *(previously section 3.2)*

**4 Discussion and conclusions**

**Appendices**
    Appendix A: Eccentricity equation correction
    Appendix B: Sea surface temperatures
    Appendix C: Impact of different initial ice sheets *(previously section 3.3)*
    Appendix D: Wave 2 methodology
    Appendix E: Metrics vs parameters plots

We now provide a point-by-point response to the reviewers' comments.

**Response to Reviewer #1**

"Patterson et al. present coupled ice sheet – climate simulations of the last two glacial maxima (last and penultimate glacial maxima, LGM and PGM). They run an ensemble of simulations varying model parameters for the LGM and the PGM. The methodology follows what their group has done in several previous papers for the LGM. From the large ensemble, Patterson et al. keep only the members that match the estimates of LGM ice sheet volume and extent. They then discuss the simulated ice sheets at the PGM given the LGM constraints.

The fact of using coupled ice sheet – climate model with a fast GCM is certainly an improvement with respect to most of the existing literature. However I am not sure that the methodology chosen is the best approach to study PGM and LGM differences. It is unclear to me so far what the main message of the paper should be since I do not currently really see what is the major new finding. I provide my comments below."

➢ We thank the reviewer for their constructive and helpful comments, which we have used to improve our manuscript.

Major comments

**RC1.1: "**It seems that the major conclusion of this study is that there is a very strong sensitivity of the simulated ice sheets to the initial ice sheet configuration. This is also what stands out from the abstract. This is an unsurprising result given the fact that the feedbacks linked to albedo (surface mass balance and temperature) are one of the most influential for climate and ice sheet dynamics. Although unsurprising I agree that it is worth studying and documenting. However this point first appears in Sec. 3.4, so relatively late in the manuscript, and as a sensitivity analysis. I would have expected more analysis and discussion on these simulations. For example: spatial pattern of ice gain/loss under PGM/LGM conditions, using the same ice sheet boundary conditions? Where these spatial pattern differences come from?…

➢ The spatial patterns seen in Figure 10 follow the outline of the different ice extents we used as boundary conditions in the sensitivity simulations. What we see in figure 10(a,d) is the location of ice growth in simulations that start from a smaller ice sheet and a bigger ice sheet. This makes these panels difficult to interpret. **To address this comment and make the key findings of the sensitivity experiments clearer, we will replace figure 10 with maps of final ice volumes from the four individual sensitivity simulations that use matching FAMOUS and Glimmer initial conditions in the revised section 3.2. We will also include, in the discussion, a map showing the difference between two runs that both use PGM ice sheet boundary conditions but LGM and PGM climates to more clearly show the spatial pattern of ice sheet growth at the PGM as a result of PGM climate. We will also add a table of the ice volumes in each simulation.**

I have the feeling that it would have been much more informative to reshape the manuscript in order to present the experiments with identical ice sheet boundary conditions as the main results. In this way the respective impact of climate and ice sheet changes, the most interesting thing of this paper, could have been presented more clearly and thoroughly."

> **We will do this (see new structure and main change 2 on page 1), rebalancing the manuscript to better highlight the results of original section 3.4 and include further analysis of these simulations.**

RC1.2: "There is one thing that is not very clear to me in the experimental setup. The climate model uses PMIP4 ice sheet boundary conditions representative of LGM and PGM, as explained from L138. For Glimmer, the initial ice sheets (North America and Greenland) are the one at 18.2 ka BP from a previous experiments. Since all the simulations are bi-directionnally coupled as shown in Fig. 1 it means that the climate model in fact does not use the PMIP4 boundary conditions for North America and Greenland but use the 18.2 ka BP ice sheets. Is this correct? If yes, where does the difference in albedo discussed in Sec. 3.3 come from?"

> During the coupling between FAMOUS and Glimmer, FAMOUS 3D ice fractions are only changed incrementally based on the ice sheet changes between two coupling time steps. When initial conditions are different in FAMOUS and Glimmer (as in our ensemble of Last Glacial Maximum, LGM, simulations), the FAMOUS ice extent over the North American continent is not updated to match the Glimmer initial conditions. Thus, in our ensemble of LGM simulations, the albedo remains high throughout Canada because the FAMOUS ice extent remains as large as the FAMOUS initial condition (Glac1D reconstruction). The implications for the ensemble are outlined in original section 3.3, which shows the difference in surface mass balance (SMB) between the LGM and Penultimate Glacial Maximum (PGM) simulations due to these different FAMOUS initial conditions causing different albedos. When FAMOUS and Glimmer start with the same ice sheet extent (as is the case for experiments $dV_2$), the ice fractions in FAMOUS are updated at each coupling time step to match those in Glimmer.

> **We will expand our Methods section to more clearly describe this coupling procedure and the model's behavior when different initial conditions are used in the climate and ice sheet models. We will include a table outlining the initial ice sheets used in each simulation and a plot of the FAMOUS ice sheet boundary conditions.**

RC1.3: "Sec. 3.1 is mostly methodological, which is fine, but I don't see what can we learn from it. The best ensemble members for the LGM produce also realistic PGM ice sheets, but the fact the PGM ice sheets are smaller are linked to the chosen ice sheet boundary conditions as shown later…

> There were two aims of running ensembles of LGM and PGM simulations described in Sec 3.1:
>> o (1) to find a combination of uncertain input parameter values that produced realistic simulations of the LGM and PGM; We used the best ensemble member

to perform the analysis on the relative impact of initial condition and climate. **We will clarify this in revised section 2.5.**

- o (2) to analyse how uncertain model parameters affect the differences in ice volume between the LGM and PGM (This is presented in original section 3.2). **In revised section 3.3, we will acknowledge that although the majority of the difference in ice volume is a result of the chosen boundary conditions, it is still useful to analyse the effect that uncertain model parameters have on this result and see whether both periods display similar sensitivities.**

In Sec. 3.2 we have not enough ensemble members to draw any conclusion I think. The only parameters that show some clear impacts are Rho and AV_GR. For the other parameters, there are good ensemble members that span the whole range. So here again I am not sure what conclusions can be drawn for the reader.

- ➢ The size of our ensemble is indeed limited compared to the study of Gandy et al. and we find that some of our conclusions overlap their findings. However, we are able to show that uncertain model parameters have only a very small impact on the difference in ice sheet volume between the LGM and PGM, which is the key finding here.
- ➢ **We will clarify this finding in our revised manuscript.**

Sec 3.3 does not explain anything. Basically it is said that the LGM as a higher albedo in the saddle region which explains the more positive SMB. But why is that (see my point 2)? If this is a result of ice sheet boundary conditions (as discussed in Sec. 3.4) I have the feeling that it should be presented later.

Given this, It would have made sense I think to group together Sec. 3.1 and 3.2 and to present Sec. 3.3 after what is currently Sec. 3.4."

- ➢ See the new structure and main change 2, above. **Specifically, on this point, we will move Section 3.3 to the appendix and refer the reader to that section for more information on the implication of the coupling procedure. The coupling procedure itself will be more clearly described in the methods (see Response to RC1.2).**

RC1.4: "Sensitivity to the oceanic forcing is hardly discussed. I think it is a quite strong limitation of this study. First, there is no real justification on the fact of using HadCM3. Then, how the ice sheet evolution would be impacted by different SST?"

- ➢ Given the low resolution of the FAMOUS model, using a dynamical ocean can introduce large biases in the simulated climate. By prescribing SSTs, we are able to limit the amplification of climate biases in this first paired set of coupled climate-ice simulations of the LGM and PGM. Ideally, we would have a set of statistically varied reconstructions to cover uncertain SST inputs (as in Gandy et al., 2023). However, pragmatically, due to the lack of both empirical and modelled PGM SST data available, we were unable to produce an equivalent PGM reconstruction. We wanted to choose PGM and LGM SST inputs consistent with each other, but there were no simulations of PGM SSTs from FAMOUS. We therefore chose to use SSTs simulated by the HadCM3 model because FAMOUS and HadCM3 share the same physics and mainly differ only in their resolution;

indeed, HadCM3 is the parent tuning target for FAMOUS (Smith et al., 2008). Further work will be required to investigate the feedbacks between ice sheets and sea surface at the PGM, but this is beyond the scope of this study.

➢ **We will include this explanation of our rationale for the choice of SST forcing in the methods. The limitations of this choice, the effect of using the different SSTs or an interactive ocean will be discussed in the discussions and conclusions section.**

**RC1.5:** "There is insufficient background to be able to link the parameters values to actual SMB changes. Since the equations are not shown and very little description is given we don't know how the different parameters play in the model."

➢ **We will add more details in the description of Table 2 and cite Smith et al., 2021, where the equations and complete model description can be found.**

**RC1.6:** "In the introduction there is no review of published works using ice sheet – climate models. There are now a relatively large literature for glacial inceptions, glacial terminations or the whole cycle with various climate models of intermediate complexity (CLIMBER-2, CLIMBER-X, LOVECLIM, iLOVECLIM, BERN3D, UVic). They have generally documented the impact of initial ice sheet configurations and the importance of albedo for ice sheet evolution. As such, I think they deserve at least a dedicated paragraph to understand how this paper is participating to knowledge increment with respect to this literature."

➢ **We will add the suggested paragraph to the introduction, highlighting these studies, their findings and limitations and how the use of a GCM improves on this.**

Minor comments / questions

"L13-14. It is a subjective but strong statement. The ice sheet response to different insolation/GHG pathways through the last two deglaciations might also be "crucial"..."

➢ **We have removed this sentence in response to a comment from Reviewer #2**

"L48-49. I don't think this is a strong constraint given the uncertainties in term of timing of the maximal extent of the Eurasian ice sheet."

➢ The purpose here is to document transparently the evidence/estimates from previous studies as examples of the difference in NH ice sheet configuration between the LGM and PGM. The uncertainty is reflected in the large range of values accepted.

➢ **We will add a sentence to clarify that the timing of the maximum extent of the EIS at the LGM is also uncertain and areas of the ice margin likely reached their maximum extents at different times throughout the glacial cycle.**

"L61-70. I would suggest to remove this part as it has very little link to the general purpose of the study. In addition, the differences listed here might be linked to differing ice sheet and climate configurations at the glacial maxima but they are also most likely linked to different insolation evolutions."

➢ **We will remove this part**

"L112. Unclear. 50 decades of climate years, meaning 500 years simulated per day? Seems very quick for a low-res GCM."

➢ Each simulation was run using 8 processors so this is ~192 core hours.
➢ **We will add this to the text.**

"Fig. 1. Explain better what is represented. Horizontal line in top left map? Solid line in bottom left graph?"

➢ **We will update the figure caption with a complete description.**

"L122-123. Strictly speaking you do not follow the PMIP4 protocols since you use interactive ice sheets that overwrite the ice sheets (as shown in Fig. 1). Also Menviel et al. (2019) present a protocol for deglaciation with prescribed ice sheet."

➢ **We will change the wording to clarify that the PMIP4 ice sheets are only used for the prescribed Antarctic and Eurasian ice sheets. However, note that the ice fractions from the PMIP4 North American Ice Sheet are also used by the climate model (see response to RC1.2).**

"Table 1. You should add a column with the reference for the ice sheets. "

➢ **We will add this column**

"L138-157. Please show the ice sheet boundary conditions used in the climate model (including over Eurasia) for the PGM and LGM."

➢ **We will add in a figure of the LGM and PGM topography anomaly from present day as implemented in the FAMOUS model in revised section 2.2.2.**

"L145. "constant" but with a seasonal cycle right? Daily forcing?"

➢ The constant forcing is composed of a complete annual cycle (including seasonality) at monthly resolution. **We will clarify this in the revised text.**

"L146. Reference for these simulations?"

➢ These are from Paul Valdes personal communication; **we will mention this in the text and include more description of how this data was produced.**

"Why these forcings and not FAMOUS computed SST and sea ice for consistency?"

➢ **See response to RC1.4**

"L148. Show summer SST as well since it seems important."

➢ **We will add this to figure 2**

"L172-173. Show difference between HadCM3 SST and reconstructions?"

➢ **We will add this to Appendix B**

"Table 2. Rho seems to be Fsnow in Gandy et al. (2023). Be consistent (at least in the paper)."

➢ **We will change to all mention of 'Rho' to 'Fsnow'**

"Table 2. Description is generally too vague. For most parameters we cannot guess in which direction the parameters can influence the simulated climate, SMB or ice sheets."

➢ **See response to RC1.5**

"Table 2. Please include the range tested for each parameters."

➢ **We will add this column to the table**

"Fig. 5. Draw the 1:1 line in b and c."

➢ **We will add this to the figure**

"L258. 4 parameters are listed here, including basal sliding. While in L458 the flow factor is mentioned and not basal sliding. Why?"

➢ The parameters Daice, Fsnow, AVGR and Flow factor are the most influential in the study by Gandy et al., 2023 and so were used for the wave 2 emulation described in original Appendix C. However, in our study, Basal sliding had a much stronger correlation to ice sheet size than Flow factor.

"L258. From the plot I clearly see a tendency for Rho and AV_GR but for the two others it is much harder. For example for Daice we see good ensemble members on both side of the tested range. For basal sliding there might be a tendency but given the fact that we don't have a lot of ensemble members here I do not think that we draw any strong statement."

➢ The analysis presented in this study is based on the correlation of ice volume and area to each parameter. These four parameters had the strongest correlation (> 0.3) with AVGR and Daice being the strongest and Basal sliding being particularly strong for ice volume. However, we acknowledge in our manuscript that "'Due to the sampling strategy, this ensemble is not the best design to analyse the sensitivity of the ice sheets during the two time periods to the different parameters and would require a larger ensemble and a sensitivity analysis with Gaussian Process emulation (e.g. Pollard et al., 2023), as is presented in Gandy et al. (2023) and Sherriff-Tadano et al. (2023). "

➢ **We will clarify in the revised manuscript that this analysis is based on the correlation**

"L267. There is something unclear, I guess in the representation. On Fig. D3 we see ice volume difference of -1 to 1 e7 km3 which seems not minor with respect to the ice volume of about 3 e7 km3."

➢ **We will remove the x10^7 and x10^6 on the volume and area labels as this was a mistake.**

"L267-269. I don't see this result in the plot. Please clarify this."

➢ **We will add more detail.** I.e., there is a negative correlation between the difference in ice volume and area between the LGM and PGM and the parameters AV_GR, basal sliding, and RHCrit and a positive correlation to Daice. (Fig. D3). This suggests that lower values of AV_GR and higher values of Daice and thus a higher albedo, as well as lower ice sheet velocity and more cloud, make the ice sheet more sensitive to changes in radiative forcings from the orbital boundary conditions.

"Fig. 7. The good ensemble members are always on the lower hand of the reconstructions. What about the modern bias of FAMOUS-ice?"

➢ The southern extents all fall towards the lower end of the plausible range, which is a common feature seen in other simulations using a low resolution coupled climate-ice sheet model due to biases that cause a reduced stationary wave effect over this region (Ziemen et al., 2014; Sherriff-Tadano et al., 2023; Gandy et al., 2023). We will add this to the text.

➢ Also, the lobes over the Great lakes aren't usually simulated by ice sheet models. It is thought that these were short lived features difficult to capture with ice sheet models. This could be due to missing processes in the subglacial processes or ice flow or that higher resolution is needed in the climate and ice sheet model.

➢ **We will add an explanation of these limitations in the results**

"L295. "passing all reductions", what does that mean? Aging for instance? Please clarify."

➢ The ice sheet model was not fully updating the ice fractions of FAMOUS to the reduced initial ice coverage used in glimmer. See response to RC1.2 for further explanation.

➢ **We will clarify this coupling process in the methods and the implications of this on the SMB of the ice sheets are outlined in the original section 3.3 which will be moved to the Appendix.**

"L296-297. There is something I don't understand in the set-up. L152-155 it is said that the same initial ice sheet is used for the LGM and PGM in GLIMMER. Since you use a coupling as in Fig. 1 the climate model also sees the same initial ice sheet in the saddle region. Please clarify this."

➢ See response to RC1.2. **This will be clarified in the methods.**

"Table 4. Add a column with V, Vc, Vi, Vci."

➢ **We will add a separate table with this information in the revised section 3.2**

"Table 4. FAMOUS initial ice sheet is not GLIMMER initial ice sheet? You are talking about the ice sheets outside the GLIMMER region (Eurasia)? Unclear."

➢ Due to the coupling procedure not updating the ice fractions in FAMOUS, the initial ice sheet used in FAMOUS is what will affect the ice cover at the start of the simulations, so this is the variable we have changed. See our answer to point RC1.2 for more explanation.

> ➤ **We will reframe the sensitivity tests to focus on the experiments we performed with matching FAMOUS and Glimmer ice sheet extents to avoid confusion and add a column to table 4 that details the Glimmer ice sheet used in each simulation.**

"L340-348. Remove this part. I don't understand why there is this discussion here while you don't account for vegetation changes."

> ➤ **We will remove this**

"L352. This should be shown!"

> ➤ **We will add a plot of difference in spring runoff and winter snowfall between the full PGM experiment and the PGM ice sheet with LGM climate experiment.**

"L359-375. I enjoy this section but it should be in a separate discussion section."

> ➤ **We will add this to the discussion and conclusion section in the new structure outlined on Page 1.**

"L369-370. Why comparing the insolation peak of 172 and 148 ka BP to the ones of MIS4? In terms of relative timing they should be compared with the ones of MIS3 (55 and 30 ka BP)."

> ➤ We were commenting on how the growth of the LGM ice sheet seen during MIS4 may not have been able to occur during the equivalent period prior to the PGM due to the higher insolation peak.
> ➤ **We will alter the text to clarify this.**

"Fig. 11. You should use the same x-axis scale. Here there is a distortion (longer period preceding the LGM than PGM) that makes the comparison difficult to do. You could group the two cycles in one graph only."

> ➤ **We will group the two cycles onto one figure panel**

"Fig. 11. In terms of insolation in the Northern Hemisphere 21 ka BP is more comparable to 137 ka BP than 140 ka BP. You should perhaps comment on this as your results could have been slightly different if using the 137 kaBP orbital and GHG configuration."

> ➤ We have used the period of highest global ice volume that is usually considered as the PGM (140ka) so it is what people are expecting.
> ➤ **We will add a comment on this in the discussion**

"L389. Why vegetation-albedo feedback is mentioned here since it is not tackled here?"

> ➤ **We will remove this**

"L390. I think it is not necessarily true. It is just that the initial ice sheet configuration is more important."

> ➤ **We will change this to read: 'In this study, the climate of each glacial maxima period has a significantly weaker effect on the simulated ice volume.'**

"Fig. A1. Relatively minor impact but with a large trend."

➤ **We will add detail on the percentage change in SMB that the corrected equation results in**

"Fig. B1. Show difference LGM-PGM as well (and summer SST)."

➤ **We have already shown the difference in Figure 2b and we will add summer SST difference to Figure 2**

"Fig. B1. The colour scale is not appropriate (SST of -20 degreeC are relatively rare)."

➤ **We will update the colour scale**

"L447. Average SMB over the ice sheet?"

➤ Yes, it is the 20-yr averaged SMB value over the ice sheet. **We will add this clarification to the text.**

"L454. Since you start from smaller ice sheets it was expected that a positive SMB was required… "

➤ Yes, the SMB needs to be positive to grow towards a LGM/PGM extent. It will need to be a bit above positive to account for mass loss to the ocean, but we set the threshold at 0 to ensure we captured all potentially reasonable simulations.

"L457-458. Be consistent with the parameters names in Tab. 2."

➤ **We will update the table**

"L486. Not observations."

➤ **We will change this to clarify that we mean empirical evidence and other model data**

"Fig. D3. Poor quality figure."

➤ **We will remake this figure**

Technical corrections

"L102. Typo, "this allows to model""

➤ We think the text is correct ('this allows it to model')

"L150. "The HadCM3 LGM SST" "

➤ **We will update the text**

"L161. Add reference of SST here."

➤ **We will add a personal communication reference in the text.**

"L170. Appendix C is mentioned before B."

> ➤ Figures B1 and B2 are mentioned in line 146

"L176,L177,L178. Set-up x 3"

> ➤ **We will update the text**

"L230 Fig. 5 has not yet been mentioned."

> ➤ **We will swap the figures**

"L263. Typo, two dots."

> ➤ **We will fix this in the text**

"L327. Keep one notation: 10**6 but not 10**7"

> ➤ **We will update the text**

"L367. Why reference to Bonelli et al. (2009) here?"

> ➤ **We will remove this reference as it is not needed**

"L420. Opening parenthesis missing."

> ➤ **We will fix this in the text**

"L420. Define nu."

> ➤ **We will define all terms in equation 4**

**Response to Reviewer #2**

"In this article, Patterson et al, perform coupled ice sheet and climate simulations. They run a wide ensemble of simulations to assess the parametric uncertainty. The subsequent analyses allows then to gain some conclusions about the effects of the different orbital configurations and initial states on the final ice sheet configurations.

Coupling a GCM to an ice sheet model is in itself of great value and informative for the community. The manuscript is very well written. The introduction adequately deals with the existing knowledge of the subject. The analysis of the results is very exhaustive and clear. And the conclusions appear generally justified with respect to what is shown in the rest of the article. Therefore, I find this work is well suited for Climate of the Past, and I recommend publication subjected to some clarifications of the experimental set up and their potential implications for the main conclusions of the study.

More specifically, I found the strategy concerning initialization a bit strange and not clearly described. Thus, my main concern is about the experimental set up and is the following:"

> ➤ We thank the reviewer for their comments which were very helpful and have helped improve our manuscript

Major comments

**RC2.1:** "The paragraph starting at lines 123 reads: "Our FAMOUS-ice simulations are set up following the Paleoclimate Modelling Intercomparison Project Phase 4 (PMIP4) protocols for the LGM (Kageyama et al., 2017) and PGM (Menviel et al., 2019)."

Around line 139:"In the climate model, the global orography (including ice sheets) and land-sea mask for the LGM are calculated from the GLAC1D 21 ka BP reconstruction (Tarasov et al., 2012) which is one of the two options in the PMIP4 protocol (Kageyama et al., 2017). For the PGM simulations we used the 140 ka BP combined ice sheet reconstruction (Tarasov et al., 2012; Abe-Ouchi et al., 2013; Briggs et al., 2014)

And line 153 reads: "In the ice sheet model, we use the same ice sheet domain and initial condition for the LGM and PGM, [...] and the initial ice sheet extent, thickness and bedrock elevation is from a previous Last Deglaciation ensemble of the NAIS, at 18.2 ka BP"

So the reader can easily wonder why using different initial conditions for the ice–sheet and the climate models. It is not clear whether this is the best way to address the influence of the initial conditions on the final ice sheet configurations, as stated in the abstract and conclusions."

  ➢ Our choice of initial conditions for the climate and ice sheet components of the coupled model was influenced by technical challenges and pragmatism. For FAMOUS, using the PMIP4 experimental design to create our boundary conditions was the most straightforward choice and presented the advantage of allowing us to compare our results with other PMIP4 simulations. The PMIP4 boundary conditions are designed for climate models and did not include the data needed to initialise an ice sheet model (e.g. bedrock elevation, ice thickness and ice temperatures). When running our ensembles of simulations, we chose to use the same initial ice sheet condition as Gandy et al. so that we could make use of the results of their large ensembles of LGM simulations. We would not claim that our approach is "the best way to address the influence of the initial conditions on the final ice sheet configurations", but they are a pragmatic way forwards, and as the first simulations of the PGM with a complex coupled climate-ice sheet model, we think they do offer valuable insight into climate-ice sheet interactions at the PGM, paving the way for further study.

**RC2.2:** "Reciprocally, concluding that the climate boundary conditions, if considered in isolation, imply a larger PGM might be dependent on the way the ice sheet initial conditions are managed under the current experimental set up. In other words, if the ice sheet model was initialized with an ice sheet configuration close to the PGM reconstruction (which, as far as I understood, has been used by the climate model as a boundary condition) it is conceivable that the climate does not react in the same manner than using a 18.2 kyr reconstruction, so that at the end, both the climate and the final ice sheet configurations widely differ with respect to what has been concluded here."

  ➢ To clarify, we performed two sets of sensitivity experiments described in the original section 3.4. The first set ($dV_1$) use the same initial 18.2 ka ice sheet in Glimmer, but different climate model ice sheets. However, the second set ($dV_2$) use matching ice

sheets in FAMOUS and Glimmer for both the LGM and PGM initial ice conditions. The results from both of these sensitivity experiments (Figure 10) display similar results, demonstrating that the climate reacts in a similar way whether the ice sheet model was initialised with a PGM reconstruction or 18.2 ka reconstruction.

➢ **We will restructure the manuscript to focus on the results of the sensitivity runs performed with matching ice sheets for the climate and ice sheet model (see new structure outlined above) and will insert a table outlining the different ice sheets used in each model for each experiment to make this clearer.**

**RC2.3:** "As a modeler, I am aware that there is not a perfect strategy for initializing the ice sheet model when the focus is on two single time snapshots. It is understandable then that using a previous deglaciation run at 18.2 kyrs has the advantage that the temperature profiles and thus viscosity have at least some internal consistency."

➢ Yes, this gets to the crux of our challenge. The need to initialise our simulations with a spun-up temperature profile was one of the reasons Gandy et al. chose a mid-deglaciation as their initial condition and we needed to use the same initial conditions in order to utilise their work in this advancement.

**RC2.4:** "However, someone could also wonder why not initializing with the ice sheet configurations that have been used as boundary conditions for the climate model (particularly so if SSTs and sea ice are fixed). You could then let the ice sheet model run to achieve internal equilibrium with the initial climate for several thousand years and subsequently "liberate" the coupled system and see where it goes.

If you have done something in these lines, I recommend incorporating it into the manuscript. If not, and you consider this suggestion unfeasible or out of the scope, please state why (there might be some subtle technical arguments I am not considering)."

➢ See response to RC2.1

**RC2.5:** "I would still encourage the authors to include a discussion on how the choices of the initialization of the experimental set up could alter the main findings of the current study."

➢ **We will add this discussion to our manuscript.**

Minor/technical comments:

"Why using Tarasov's reconstruction for the LGM and the combined one for the PGM?"

➢ These are the ice sheet configurations specified in the respective LGM and PGM PMIP4 protocols. The LGM has a choice of 3 configurations whereas the PGM only has the one combined forcing.

"Lines 14 and 15 of the abstract read: "Therefore, a better understanding of how and why these two glacial maxima differed is crucial for developing the full picture on why the Last Interglacial sea level was up to 9 meters higher than today, and thus may help constrain future sea level rise.""

This makes sense but is not addressed at all in the rest of the manuscript. Therefore, I suggest removing it or elaborate something in the discussion on the potential implications of your findings on this matter."

➢ **We will remove this line**

---

## Author Response (AR1)

**Response to reviews of 'Patterson et al., Contrasting the Penultimate and Last Glacial Maxima (140 and 21 ka BP) using coupled climate-ice sheet modelling'**

We thank both reviewers for their constructive and helpful comments, which we have used to improve our manuscript. We were happy to hear that the reviewers think our use of coupled climate-ice sheet modelling is an "improvement on existing literature" and "of great value and informative for the community". Following the reviewers' suggestions, we propose the following main changes:

1. Better description of our coupling procedure and clearer justification of the choices of initial and boundary conditions.

2. Adjusting the manuscript's outline to make the results from our factorial decomposition (original section 3.4 Sensitivity Analysis) the central feature of our manuscript. This includes moving some text from the results sections to the methods (new sections 2.4 and 2.5), moving old section "3.3 Climate-ice sheet interactions" to the methods and appendix and moving the section "Uncertainty due to model parameters" after the main results section "3.2 Impact of initial ice sheet vs climate". The new structure of the manuscript will be as follows:

**1 Introduction**

**2 Methods**

2.1 Model description

2.2 Experiment design

    2.2.1 Climate boundary conditions

    2.2.2 Ice sheet boundary and initial conditions *(New section)*

2.3 Ensemble design

2.4 Implausibility criteria *(methods text moved from section 3.1.2)*

2.5 Sensitivity analysis *(methods text moved from section 3.4)*

**3 Results**

3.1 Ensembles

3.2 Impact of initial ice sheet vs climate *(previously results from section 3.4)*

3.3 Uncertainty due to model parameters *(previously section 3.2)*

**4 Discussion**

**5 Conclusions**

**Appendices**

Appendix A: Eccentricity equation correction

Appendix B: Sea surface temperatures

Appendix C: Impact of different initial ice sheets *(New section)*

Appendix D: Wave 2 methodology

Appendix E: Metrics vs parameters plots

We now provide a point-by-point response to the reviewers' comments.

**Response to Reviewer #1**

"Patterson et al. present coupled ice sheet – climate simulations of the last two glacial maxima (last and penultimate glacial maxima, LGM and PGM). They run an ensemble of simulations varying model parameters for the LGM and the PGM. The methodology follows what their group has done in several previous papers for the LGM. From the large ensemble, Patterson et al. keep only the members that match the estimates of LGM ice sheet volume and extent. They then discuss the simulated ice sheets at the PGM given the LGM constraints.

The fact of using coupled ice sheet – climate model with a fast GCM is certainly an improvement with respect to most of the existing literature. However I am not sure that the methodology chosen is the best approach to study PGM and LGM differences. It is unclear to me so far what the main message of the paper should be since I do not currently really see what is the major new finding. I provide my comments below."

➢ We thank the reviewer for their constructive and helpful comments, which we have used to improve our manuscript.

**Major comments**

RC1.1: "It seems that the major conclusion of this study is that there is a very strong sensitivity of the simulated ice sheets to the initial ice sheet configuration. This is also what stands out from the abstract. This is an unsurprising result given the fact that the feedbacks linked to albedo (surface mass balance and temperature) are one of the most influential for climate and ice sheet dynamics. Although unsurprising I agree that it is worth studying and documenting. However this point first appears in Sec. 3.4, so relatively late in the manuscript, and as a sensitivity analysis. I would have expected more analysis and discussion on these simulations. For example: spatial pattern of ice gain/loss under PGM/LGM conditions, using the same ice sheet boundary conditions? Where these spatial pattern differences come from?…

> ➤ The spatial patterns seen in Figure 10 follow the outline of the different ice extents we used as boundary conditions in the sensitivity simulations. What we see in figure 10(a,d) is the location of ice growth in simulations that start from a smaller ice sheet and a bigger ice sheet. This makes these panels difficult to interpret. **To address this comment and make the key findings of the sensitivity experiments clearer, we have replaced figure 10 with maps of final ice volumes from the four individual sensitivity simulations that use matching FAMOUS and Glimmer initial conditions in the revised section 3.2 (new figure 8, L361). We have also included a map showing the difference between two runs that both use PGM ice sheet boundary conditions but LGM and PGM climates to more clearly show the spatial pattern of ice sheet growth at the PGM as a result of PGM climate (new figure 9, L371). We have also added a table of the ice volumes in each simulation (new Table 5, L351).**

I have the feeling that it would have been much more informative to reshape the manuscript in order to present the experiments with identical ice sheet boundary conditions as the main results. In this way the respective impact of climate and ice sheet changes, the most interesting thing of this paper, could have been presented more clearly and thoroughly."

➢ We have done this (see new structure and main change 2 on page 1), rebalancing the manuscript to better highlight the results of original section 3.4 and include further analysis of these simulations. See revised section 3.2.

RC1.2: "There is one thing that is not very clear to me in the experimental setup. The climate model uses PMIP4 ice sheet boundary conditions representative of LGM and PGM, as explained from L138. For Glimmer, the initial ice sheets (North America and Greenland) are the one at 18.2 ka BP from a previous experiments. Since all the simulations are bi-directionnally coupled as shown in Fig. 1 it means that the climate model in fact does not use the PMIP4 boundary conditions for North America and Greenland but use the 18.2 ka BP ice sheets. Is this correct? If yes, where does the difference in albedo discussed in Sec. 3.3 come from?"

➢ During the coupling between FAMOUS and Glimmer, FAMOUS 3D ice fractions are only changed incrementally based on the ice sheet changes between two coupling time steps. When initial conditions are different in FAMOUS and Glimmer (as in our ensemble of Last Glacial Maximum, LGM, simulations), the FAMOUS ice extent over the North American continent is not updated to match the Glimmer initial conditions. Thus, in our ensemble of LGM simulations, the albedo remains high throughout Canada because the FAMOUS ice extent remains as large as the FAMOUS initial condition (Glac1D reconstruction). The implications for the ensemble are outlined in original section 3.3, which shows the difference in surface mass balance (SMB) between the LGM and Penultimate Glacial Maximum (PGM) simulations due to these different FAMOUS initial conditions causing different albedos. When FAMOUS and Glimmer start with the same ice sheet extent (as is the case for experiments $dV_2$), the ice fractions in FAMOUS are updated at each coupling time step to match those in Glimmer.

➢ We have expanded our Methods section to more clearly describe this coupling procedure and the model's behavior when different initial conditions are used in the climate and ice sheet models (new section 2.2.2, L188-198). We have included a table outlining the initial ice sheets used in each simulation (new Table 2, L210) and a plot of the FAMOUS ice sheet boundary conditions (new figure 3, L201).

**RC1.3**: "Sec. 3.1 is mostly methodological, which is fine, but I don't see what can we learn from it. The best ensemble members for the LGM produce also realistic PGM ice sheets, but the fact the PGM ice sheets are smaller are linked to the chosen ice sheet boundary conditions as shown later…

- ➤ There were two aims of running ensembles of LGM and PGM simulations described in Sec 3.1:
    - ○ (1) to find a combination of uncertain input parameter values that produced realistic simulations of the LGM and PGM; We used the best ensemble member to perform the analysis on the relative impact of initial condition and climate. **We have clarified this in revised section 2.5.**
    - ○ (2) to analyse how uncertain model parameters affect the differences in ice volume between the LGM and PGM (This is presented in original section 3.2 and **revised section 3.3**).

In Sec. 3.2 we have not enough ensemble members to draw any conclusion I think. The only parameters that show some clear impacts are Rho and AV_GR. For the other parameters, there are good ensemble members that span the whole range. So here again I am not sure what conclusions can be drawn for the reader.

- ➤ The size of our ensemble is indeed limited compared to the study of Gandy et al. and we find that some of our conclusions overlap their findings. However, we are able to show that uncertain model parameters have only a very small impact on the difference in ice sheet volume between the LGM and PGM, which is the key finding here.
- ➤ **We have clarified this finding in our revised manuscript (revised section 3.3).**

Sec 3.3 does not explain anything. Basically it is said that the LGM as a higher albedo in the saddle region which explains the more positive SMB. But why is that (see my point 2)? If this is a result of ice sheet boundary conditions (as discussed in Sec. 3.4) I have the feeling that it should be presented later.

Given this, It would have made sense I think to group together Sec. 3.1 and 3.2 and to present Sec. 3.3 after what is currently Sec. 3.4."

➢ See the new structure and main change 2, above. **Specifically, on this point, we have removed original Section 3.3 and explained in new section 2.2.2 the implication of the coupling procedure and the impact of the different initial ice sheets is explored in revised section 3.2 (see Response to RC1.2).**

RC1.4: "Sensitivity to the oceanic forcing is hardly discussed. I think it is a quite strong limitation of this study. First, there is no real justification on the fact of using HadCM3. Then, how the ice sheet evolution would be impacted by different SST?"

➢ Given the low resolution of the FAMOUS model, using a dynamical ocean can introduce large biases in the simulated climate. By prescribing SSTs, we are able to limit the amplification of climate biases in this first paired set of coupled climate-ice simulations of the LGM and PGM. Ideally, we would have a set of statistically varied reconstructions to cover uncertain SST inputs (as in Gandy et al., 2023). However, pragmatically, due to the lack of both empirical and modelled PGM SST data available, we were unable to produce an equivalent PGM reconstruction. We wanted to choose PGM and LGM SST inputs consistent with each other, but there were no simulations of PGM SSTs from FAMOUS. We therefore chose to use SSTs simulated by the HadCM3 model because FAMOUS and HadCM3 share the same physics and mainly differ only in their resolution; indeed, HadCM3 is the parent tuning target for FAMOUS (Smith et al., 2008). Further work will be required to investigate the feedbacks between ice sheets and sea surface at the PGM, but this is beyond the scope of this study.

➢ **We include this explanation of our rationale for the choice of SST forcing in the methods (L153-175). The limitations of this choice, the effect of using the different SSTs or an interactive ocean is discussed in the discussion section (L471-486).**

**RC1.5:** "There is insufficient background to be able to link the parameters values to actual SMB changes. Since the equations are not shown and very little description is given we don't know how the different parameters play in the model."

➢ We have added more details in the description of new Table 3 and cite Smith et al., 2021, where the equations and complete model description can be found.

**RC1.6:** "In the introduction there is no review of published works using ice sheet – climate models. There are now a relatively large literature for glacial inceptions, glacial terminations or the whole cycle with various climate models of intermediate complexity (CLIMBER-2, CLIMBER-X, LOVECLIM, iLOVECLIM, BERN3D, UVic). They have generally documented the impact of initial ice sheet configurations and the importance of albedo for ice sheet evolution. As such, I think they deserve at least a dedicated paragraph to understand how this paper is participating to knowledge increment with respect to this literature."

➢ We have added the suggested paragraph to the introduction, highlighting these studies, their findings and limitations and how the use of a GCM improves on this (L67-80).

Minor comments / questions

"L13-14. It is a subjective but strong statement. The ice sheet response to different insolation/GHG pathways through the last two deglaciations might also be "crucial"..."

➢ We have removed this sentence in response to a comment from Reviewer #2

"L48-49. I don't think this is a strong constraint given the uncertainties in term of timing of the maximal extent of the Eurasian ice sheet."

➢ The purpose here is to document transparently the evidence/estimates from previous studies as examples of the difference in NH ice sheet configuration between the LGM and PGM. The uncertainty is reflected in the large range of values accepted.

➢ We have added a sentence to clarify that the timing of the maximum extent of the EIS at the LGM is also uncertain and areas of the ice margin likely reached their maximum extents at different times throughout the glacial cycle (L39-41).

"L61-70. I would suggest to remove this part as it has very little link to the general purpose of the study. In addition, the differences listed here might be linked to differing ice sheet and climate configurations at the glacial maxima but they are also most likely linked to different insolation evolutions."

➢ We have removed this part

"L112. Unclear. 50 decades of climate years, meaning 500 years simulated per day? Seems very quick for a low-res GCM."

➢ Each simulation was run using 8 processors so this is ~192 core hours.
➢ We have added this to the text (L121).

"Fig. 1. Explain better what is represented. Horizontal line in top left map? Solid line in bottom left graph?"

➢ We have updated the figure caption with a complete description (L118).

"L122-123. Strictly speaking you do not follow the PMIP4 protocols since you use interactive ice sheets that overwrite the ice sheets (as shown in Fig. 1). Also Menviel et al. (2019) present a protocol for deglaciation with prescribed ice sheet."

➢ We have changed the wording to clarify that the PMIP4 ice sheets are only used for the prescribed Antarctic and Eurasian ice sheets (L128 and 145). However, note that the ice fractions from the PMIP4 North American Ice Sheet are also used by the climate model (see response to RC1.2).

"Table 1. You should add a column with the reference for the ice sheets. "

➢ We have added this column

"L138-157. Please show the ice sheet boundary conditions used in the climate model (including over Eurasia) for the PGM and LGM."

➤ We have added a figure of the LGM and PGM topography anomaly from present day as implemented in the FAMOUS model in revised section 2.2.2 (new Figure 3).

"L145. "constant" but with a seasonal cycle right? Daily forcing?"

➤ The constant forcing is composed of a complete annual cycle (including seasonality) at monthly resolution. **We have clarified this in the revised text (L169-171).**

"L146. Reference for these simulations?"

➤ We have added a reference and more details for the LGM and PGM simulations (L156-163)

➤ We have also added Paul Valdes as a coauthor as he provided this data and this information

"Why these forcings and not FAMOUS computed SST and sea ice for consistency?"

➤ See response to RC1.4

"L148. Show summer SST as well since it seems important."

➤ We have added this to figure 2

"L172-173. Show difference between HadCM3 SST and reconstructions?"

➤ We have added this to Appendix B (Figure B1c)

"Table 2. Rho seems to be Fsnow in Gandy et al. (2023). Be consistent (at least in the paper)."

➤ We have changed all mention of 'Rho' to 'Fsnow'

"Table 2. Description is generally too vague. For most parameters we cannot guess in which direction the parameters can influence the simulated climate, SMB or ice sheets."

➢ See response to RC1.5

"Table 2. Please include the range tested for each parameters."

➢ We have added this column to the table (new Table 3)

"Fig. 5. Draw the 1:1 line in b and c."

➢ We have added this to the figure (new Figure 6)

"L258. 4 parameters are listed here, including basal sliding. While in L458 the flow factor is mentioned and not basal sliding. Why?"

➢ The parameters Daice, Fsnow, AVGR and Flow factor are the most influential in the study by Gandy et al., 2023 and so were used for the wave 2 emulation described in original Appendix C. However, in our study, Basal sliding had a much stronger correlation to ice sheet size than Flow factor.

"L258. From the plot I clearly see a tendency for Rho and AV_GR but for the two others it is much harder. For example for Daice we see good ensemble members on both side of the tested range. For basal sliding there might be a tendency but given the fact that we don't have a lot of ensemble members here I do not think that we draw any strong statement."

➢ The analysis presented in this study is based on the correlation of ice volume and area to each parameter. These four parameters had the strongest correlation (> 0.3) with AVGR and Daice being the strongest and Basal sliding being particularly strong for ice volume. However, we acknowledge in our manuscript that "'Due to the sampling strategy, this ensemble is not the best design to analyse the sensitivity of the ice sheets during the two time periods to the different parameters and would require a larger ensemble and a sensitivity analysis with Gaussian Process emulation (e.g. Pollard et al., 2023), as is presented in Gandy et al. (2023) and Sherriff-Tadano et al. (2023). " We have emphasised this in L376-379

➢ We have clarified in the revised manuscript that this analysis is based on the correlation (L381)

"L267. There is something unclear, I guess in the representation. On Fig. D3 we see ice volume difference of -1 to 1 e7 km3 which seems not minor with respect to the ice volume of about 3 e7 km3."

> We have removed the x10^7 and x10^6 on the volume and area labels as this was a mistake.

"L267-269. I don't see this result in the plot. Please clarify this."

> We have added more detail (L389-393). I.e., there is a negative correlation between the difference in ice volume and area between the LGM and PGM and the parameters AV_GR, basal sliding, and RHCrit and a positive correlation to Daice. (Fig. D3). This suggests that lower values of AV_GR and higher values of Daice and thus a higher albedo, as well as lower ice sheet velocity and more cloud, make the ice sheet more sensitive to changes in radiative forcings from the orbital boundary conditions.

"Fig. 7. The good ensemble members are always on the lower hand of the reconstructions. What about the modern bias of FAMOUS-ice?"

> The southern extents all fall towards the lower end of the plausible range, which is a common feature seen in other simulations using a low resolution coupled climate-ice sheet model due to biases that cause a reduced stationary wave effect over this region (Ziemen et al., 2014; Sherriff-Tadano et al., 2023; Gandy et al., 2023). We will add this to the text.

> Also, the lobes over the Great lakes aren't usually simulated by ice sheet models. It is thought that these were short lived features difficult to capture with ice sheet models. This could be due to missing processes in the subglacial processes or ice flow or that higher resolution is needed in the climate and ice sheet model.

> We have highlighted this in the results (L314-317) and added an explanation of these limitations in the discussion (L403-408)

"L295. "passing all reductions", what does that mean? Aging for instance? Please clarify."

- The ice sheet model was not fully updating the ice fractions of FAMOUS to the reduced initial ice coverage used in glimmer. See response to RC1.2 for further explanation.
- We have clarified this coupling process in the methods (new section 2.2.2) and the implications of this are explored in revised section 3.2 and Appendix C.

"L296-297. There is something I don't understand in the set-up. L152-155 it is said that the same initial ice sheet is used for the LGM and PGM in GLIMMER. Since you use a coupling as in Fig. 1 the climate model also sees the same initial ice sheet in the saddle region. Please clarify this."

- See response to RC1.2. This is clarified in the methods.

"Table 4. Add a column with V, Vc, Vi, Vci."

- We have added all experiments to new Table 2 (L211)

"Table 4. FAMOUS initial ice sheet is not GLIMMER initial ice sheet? You are talking about the ice sheets outside the GLIMMER region (Eurasia)? Unclear."

- Due to the coupling procedure not updating the ice fractions in FAMOUS, the initial ice sheet used in FAMOUS is what will affect the ice cover at the start of the simulations, so this is the variable we have changed. See our answer to point RC1.2 for more explanation.
- We have reframed the sensitivity tests to focus on the experiments we performed with matching FAMOUS and Glimmer ice sheet extents to avoid confusion (revised section 3.2) and added a new Table 2 that details the initial ice sheets used in each simulation.

"L340-348. Remove this part. I don't understand why there is this discussion here while you don't account for vegetation changes."

- We have removed this

"L352. This should be shown!"

➢ We have added a plot of difference in spring runoff and winter snowfall between the full PGM experiment and the PGM ice sheet with LGM climate experiment (new Figure 9, L371).

"L359-375. I enjoy this section but it should be in a separate discussion section."

➢ We have added this to the discussion section in the new structure outlined on Page 1 (L435-452).

"L369-370. Why comparing the insolation peak of 172 and 148 ka BP to the ones of MIS4? In terms of relative timing they should be compared with the ones of MIS3 (55 and 30 ka BP)."

➢ We were commenting on how the growth of the LGM ice sheet seen during MIS4 may not have been able to occur during the equivalent period prior to the PGM due to the higher insolation peak.

➢ We have added clarification to the text (L446-449).

"Fig. 11. You should use the same x-axis scale. Here there is a distortion (longer period preceding the LGM than PGM) that makes the comparison difficult to do. You could group the two cycles in one graph only."

➢ We have grouped the two cycles onto one figure panel (figure 11)

"Fig. 11. In terms of insolation in the Northern Hemisphere 21 ka BP is more comparable to 137 ka BP than 140 ka BP. You should perhaps comment on this as your results could have been slightly different if using the 137 kaBP orbital and GHG configuration."

➢ We have used the period of highest global ice volume that is usually considered as the PGM (140ka) so it is what people are expecting.

➢ We have commented on this in the discussion (L431-433)

"L389. Why vegetation-albedo feedback is mentioned here since it is not tackled here?"

➢ We have removed this

"L390. I think it is not necessarily true. It is just that the initial ice sheet configuration is more important."

> ➤ We have reworded the conclusions (L500-513)

"Fig. A1. Relatively minor impact but with a large trend."

> ➤ We have added detail on the percentage change in SMB that the corrected equation results in (L525-528)

"Fig. B1. Show difference LGM-PGM as well (and summer SST)."

> ➤ We have already shown the difference in Figure 2b and have added summer SST difference to Figure 2

"Fig. B1. The colour scale is not appropriate (SST of -20 degreeC are relatively rare)."

> ➤ We have updated the colour scale

"L447. Average SMB over the ice sheet?"

> ➤ Yes, it is the 20-yr averaged SMB value over the ice sheet. We have added this clarification to the text (L553).

"L454. Since you start from smaller ice sheets it was expected that a positive SMB was required… "

> ➤ Yes, the SMB needs to be positive to grow towards a LGM/PGM extent. It will need to be a bit above positive to account for mass loss to the ocean, but we set the threshold at 0 to ensure we captured all potentially reasonable simulations.

"L457-458. Be consistent with the parameters names in Tab. 2."

> ➤ We have updated the table (new Table 3)

"L486. Not observations."

➢ We have changed this to clarify that we mean empirical evidence and other model data (L592)

"Fig. D3. Poor quality figure."

➢ We have remade this figure

Technical corrections

"L102. Typo, "this allows to model""

➢ We think the text is correct ('this allows it to model')

"L150. "The HadCM3 LGM SST" "

➢ We have updated the text (L174)

"L161. Add reference of SST here."

➢ We have added a reference for the LGM and PGM SSTs in the text and included a description of the PGM SSTs (L155-163).

"L170. Appendix C is mentioned before B."

➢ Figure B is mentioned in L156

"L176,L177,L178. Set-up x 3"

➢ We have updated the text (L231-233)

"L230 Fig. 5 has not yet been mentioned."

➢ We have updated the figures and ensured they are in the correct order of mention

"L263. Typo, two dots."

➢ We have fixed this

"L327. Keep one notation: 10**6 but not 10**7"

➢ We have changed the volumes to m SLE for consistency

*"L367. Why reference to Bonelli et al. (2009) here?"*

➢ We have removed this reference as it is not needed

*"L420. Opening parenthesis missing."*

➢ We have fixed this in the text (L517)

*"L420. Define nu."*

➢ We have defined all terms in equation 4 (L523-524)

**Response to Reviewer #2**

*"In this article, Patterson et al, perform coupled ice sheet and climate simulations. They run a wide ensemble of simulations to assess the parametric uncertainty. The subsequent analyses allows then to gain some conclusions about the effects of the different orbital configurations and initial states on the final ice sheet configurations.*

*Coupling a GCM to an ice sheet model is in itself of great value and informative for the community. The manuscript is very well written. The introduction adequately deals with the existing knowledge of the subject. The analysis of the results is very exhaustive and clear. And the conclusions appear generally justified with respect to what is shown in the rest of the article. Therefore, I find this work is well suited for Climate of the Past, and I recommend publication subjected to some clarifications of the experimental set up and their potential implications for the main conclusions of the study.*

*More specifically, I found the strategy concerning initialization a bit strange and not clearly described. Thus, my main concern is about the experimental set up and is the following:"*

➢ We thank the reviewer for their comments which were very helpful and have helped improve our manuscript

Major comments

**RC2.1:** "The paragraph starting at lines 123 reads: "Our FAMOUS-ice simulations are set up following the Paleoclimate Modelling Intercomparison Project Phase 4 (PMIP4) protocols for the LGM (Kageyama et al., 2017) and PGM (Menviel et al., 2019)."

Around line 139:"In the climate model, the global orography (including ice sheets) and land-sea mask for the LGM are calculated from the GLAC1D 21 ka BP reconstruction (Tarasov et al., 2012) which is one of the two options in the PMIP4 protocol (Kageyama et al., 2017). For the PGM simulations we used the 140 ka BP combined ice sheet reconstruction (Tarasov et al., 2012; Abe-Ouchi et al., 2013; Briggs et al., 2014)

And line 153 reads: "In the ice sheet model, we use the same ice sheet domain and initial condition for the LGM and PGM, [...] and the initial ice sheet extent, thickness and bedrock elevation is from a previous Last Deglaciation ensemble of the NAIS, at 18.2 ka BP"

So the reader can easily wonder why using different initial conditions for the ice–sheet and the climate models. It is not clear whether this is the best way to address the influence of the initial conditions on the final ice sheet configurations, as stated in the abstract and conclusions."

> ➤ Our choice of initial conditions for the climate and ice sheet components of the coupled model was influenced by technical challenges and pragmatism. For FAMOUS, using the PMIP4 experimental design to create our boundary conditions was the most straightforward choice and presented the advantage of allowing us to compare our results with other PMIP4 simulations. The PMIP4 boundary conditions are designed for climate models and did not include the data needed to initialise an ice sheet model (e.g. bedrock elevation, ice thickness and ice temperatures). When running our ensembles of simulations, we chose to use the same initial ice sheet condition as Gandy et al. so that we could make use of the results of their large ensembles of LGM simulations. We would not claim that our approach is "the best way to address the influence of the initial conditions on the final ice sheet configurations", but they are a pragmatic way forwards, and as the first simulations of the PGM with a complex coupled climate-ice sheet model,

we think they do offer valuable insight into climate-ice sheet interactions at the PGM, paving the way for further study.

**RC2.2:** "Reciprocally, concluding that the climate boundary conditions, if considered in isolation, imply a larger PGM might be dependent on the way the ice sheet initial conditions are managed under the current experimental set up. In other words, if the ice sheet model was initialized with an ice sheet configuration close to the PGM reconstruction (which, as far as I understood, has been used by the climate model as a boundary condition) it is conceivable that the climate does not react in the same manner than using a 18.2 kyr reconstruction, so that at the end, both the climate and the final ice sheet configurations widely differ with respect to what has been concluded here."

➢ To clarify, we performed two sets of sensitivity experiments described in the original section 3.4. The first set ($dV_1$) use the same initial 18.2 ka ice sheet in Glimmer, but different climate model ice sheets. However, the second set ($dV_2$) use matching ice sheets in FAMOUS and Glimmer for both the LGM and PGM initial ice conditions. The results from both of these sensitivity experiments (Figure 10) display similar results, demonstrating that the climate reacts in a similar way whether the ice sheet model was initialised with a PGM reconstruction or 18.2 ka reconstruction.

➢ **We have restructured the manuscript to focus on the results of the sensitivity runs performed with matching ice sheets for the climate and ice sheet model (see new structure outlined above and revised section 3.2) and will insert a table outlining the different ice sheets used in each model for each experiment to make this clearer (new Table 2).**

**RC2.3:** "As a modeler, I am aware that there is not a perfect strategy for initializing the ice sheet model when the focus is on two single time snapshots. It is understandable then that using a previous deglaciation run at 18.2 kyrs has the advantage that the temperature profiles and thus viscosity have at least some internal consistency."

> Yes, this gets to the crux of our challenge. The need to initialise our simulations with a spun-up temperature profile was one of the reasons Gandy et al. chose a mid-deglaciation as their initial condition and we needed to use the same initial conditions in order to utilise their work in this advancement.

**RC2.4:** "However, someone could also wonder why not initializing with the ice sheet configurations that have been used as boundary conditions for the climate model (particularly so if SSTs and sea ice are fixed). You could then let the ice sheet model run to achieve internal equilibrium with the initial climate for several thousand years and subsequently "liberate" the coupled system and see where it goes.

If you have done something in these lines, I recommend incorporating it into the manuscript. If not, and you consider this suggestion unfeasible or out of the scope, please state why (there might be some subtle technical arguments I am not considering)."

> See response to RC2.1

**RC2.5:** "I would still encourage the authors to include a discussion on how the choices of the initialization of the experimental set up could alter the main findings of the current study."

> **The impact of the ice sheet initialisation is explored in revised section 3.2 and Appendix C**

Minor/technical comments:

"Why using Tarasov's reconstruction for the LGM and the combined one for the PGM?"

> These are the ice sheet configurations specified in the respective LGM and PGM PMIP4 protocols. The LGM has a choice of 3 configurations whereas the PGM only has the one combined forcing.

"Lines 14 and 15 of the abstract read: "Therefore, a better understanding of how and why these two glacial maxima differed is crucial for developing the full picture on why the Last Interglacial

sea level was up to 9 meters higher than today, and thus may help constrain future sea level rise."

This makes sense but is not addressed at all in the rest of the manuscript. Therefore, I suggest removing it or elaborate something in the discussion on the potential implications of your findings on this matter."

 ➢ We have removed this line

---

## Author Response (AR2)

Responses to reviewer's comments: **Contrasting the Penultimate and Last Glacial Maxima (140 and 21 ka BP) using coupled climate-ice sheet modelling**

"First of all I would like to thank the authors for their considerable efforts in responding to my comments. Notably, the manuscript contains now some important clarifications concerning the choice of the SST forcing, the ice mask update methodology and a physical meaning of the model parameters. All these clarifications help to understand the model results. I also appreciate the stronger focus to the choice of the initial ice sheet and its impacts."

The authors would like to thank the reviewer again for the time they have put in to reviewing this manuscript and their constructive comments that have helped improve this work. We are pleased that the changes we have made have been accepted with minor revisions and have clarified the results. We have provided a point-by-point response to the minor comments below and updated the manuscript accordingly.

"I only have a few more comments:

- The specificity of the coupling that you now clarify L189-196 answers my main interrogation during my first review. I thank you for this addition. However, if I understand it correctly, this feature looks like a bug – or at least like a current model limitation. It certainly limits the ice sheet retreat for unfavourable conditions. If it is really the case, perhaps you can explicitly mention that it will be (is?) fixed in the future."

➢ **This issue has been fixed in the model and we have mentioned this in the manuscript L199-200**

"- Tab. 2 it is true that the ice thickness is not available for the PGM (as explained in L. 286) while it is for the LGM, that is why you can more easily starts for GLAC-1D reconstructions at the LGM. However, Glimmer needs presumably internal initial conditions such as temperature. Where does it come from in experiments V_1 and Vc_1?"

➢ The initial conditions that Glimmer requires are; topography, ice thickness and bedrock softness. The internal temperature evolution of the ice sheet is described by equation 19 in Rutt et al., 2009. For all simulations, the surface temperature boundary condition is set to the mean annual surface temperature up to a maximum of 0°C. The basal temperature is determined by the geothermal heat flux (which is $-5e^{-2}$) and friction from sliding. **We have added a sentence explaining this in the model description section, L109-112**.

"- L.87-88 "...this study uses a coupled climate-ice sheet model (FAMOUS...)": replace by coupled atmosphere-ice sheet model."

➢ **This has been changed**